# YouTubePD: A Multimodal Benchmark for Parkinson's Disease Analysis

**Andy Zhou**[1*]    **Samuel Li**[1*]    **Pranav Sriram**[2*]    **Xiang Li**[1*]    **Jiahua Dong**[1*]
**Ansh Sharma**[1]    **Yuanyi Zhong**[1]    **Shirui Luo**[3]    **Maria Jaromin**[3]
**Volodymyr Kindratenko**[1,2,3]    **George Heintz**[4]    **Christopher Zallek**[5]    **Yu-Xiong Wang**[1,2,3]

[1]Department of Computer Science, University of Illinois Urbana-Champaign
[2]Department of Electrical and Computer Engineering, University of Illinois Urbana-Champaign
[3]National Center for Supercomputing Applications, University of Illinois Urbana-Champaign
[4]Health Care Engineering Systems Center, University of Illinois Urbana-Champaign
[5]OSF Healthcare Illinois Neurological Institute - Neurology

https://uiuc-yuxiong-lab.github.io/YouTubePD

## Abstract

The healthcare and AI communities have witnessed a growing interest in the development of AI-assisted systems for automated diagnosis of Parkinson's Disease (PD), one of the most prevalent neurodegenerative disorders. However, the progress in this area has been significantly impeded by the absence of a unified, publicly available benchmark, which prevents comprehensive evaluation of existing PD analysis methods and the development of advanced models. This work overcomes these challenges by introducing YouTubePD – the *first* publicly available multimodal benchmark designed for PD analysis. We crowd-source existing videos featured with PD from YouTube, exploit multimodal information including *in-the-wild* videos, audios, and facial landmarks across 200+ subject videos, and provide dense and diverse annotations from a clinical expert. Based on our benchmark, we propose three challenging and complementary tasks encompassing *both discriminative and generative* tasks, along with a comprehensive set of corresponding baselines. Experimental evaluation showcases the potential of modern deep learning and computer vision techniques, in particular the generalizability of the models developed on our YouTubePD to real-world clinical settings, while revealing their limitations. We hope that our work paves the way for future research in this direction.

## 1 Introduction

As one of the most prevalent neurodegenerative disorders, Parkinson's Disease (PD) affects over 10 million people worldwide, and the number of PD subjects is projected to double within the next 20 years [2]. Notably, PD is a progressive disease, with symptoms becoming increasingly pronounced and severe over time. Meanwhile, the diagnostic and treatment costs associated with PD are substantial, amounting up to $14 billion in the United States alone [2]. Therefore, there is an urgent need for developing AI-assisted systems for automated PD assessment in the healthcare and AI communities. Such systems can facilitate the recognition of undiagnosed people with PD, aid clinicians in their evaluation, and play a crucial role in the continuing care of people with PD by monitoring disease progression and tracking responses to therapies.

However, the progress in this area has been significantly impeded by the *absence of a unified, publicly available* benchmark. For AI-driven applications, the first and foremost endeavor lies in establishing

---

*Equal contribution.

37th Conference on Neural Information Processing Systems (NeurIPS 2023) Track on Datasets and Benchmarks.

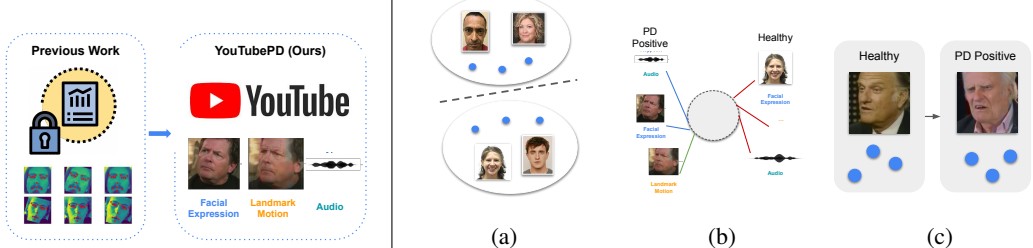

Figure 1: We propose YouTubePD, an *open-source*, *multimodal* benchmark on Parkinson's Disease (PD) analysis. Our dataset contains in-the-wild videos, audios, and facial landmarks. On the right, we show the three tasks on our benchmark: **(a)** facial-expression-based PD classification, **(b)** multimodal PD classification, and **(c)** PD progression synthesis.

a suitable benchmark. Ideally, the benchmark is openly accessible to facilitate methodology development, exemplified by the notable ImageNet benchmark that has substantially catalyzed object recognition performance [12]. In contrast, existing PD datasets are typically constructed *in clinical settings* and are thus kept private to protect patient privacy [3, 5, 17, 19, 25, 33, 41, 48]. In addition, the creation of PD datasets is a costly, time-consuming process, which requires a significant amount of effort for collecting patient data and curating annotations from qualified clinical experts.

In this paper, we overcome the aforementioned challenges by introducing an *open* benchmark that not only enables comparisons of existing PD analysis techniques, but also facilitates the development of advanced models that leverage state-of-the-art computer vision and deep learning techniques. *Our key insight* is that, instead of curating data from clinical settings as is normally the case, we exploit publicly available online resources and crowd-source existing videos featured with PD from YouTube.

These YouTube videos contain rich examination modalities, making our benchmark *inherently multimodal*. This property allows for the investigation of some particularly indicative symptoms associated with PD. More concretely, PD is often characterized by the manifestation of a motor-based symptom known as facial bradykinesia – the lessened movement of orofacial muscles and a reduction in spontaneous emotional expressions [8]. This results in a constant "mask-like" expression in PD subjects referred to as *hypomimia* [8], a sensitive biomarker for PD [1]. Following prior work [3, 5, 19, 25, 48], our benchmark considers hypomimia as a primary indicator for PD diagnosis. Moreover, our benchmark includes other symptoms experienced by PD patients, including tremors and speech changes. These additional symptoms are considered alongside hypomimia, providing a more comprehensive evaluation of the disease and capitalizing towards a multimodal solution.

**Our contribution** is described as follows:

1. To tackle the lack of a unified, open, and multimodal benchmark in PD analysis, we introduce **YouTubePD**. By leveraging YouTube videos of public figures experiencing PD, we obtain *high-quality, in-the-wild* data of Parkinson's symptoms. As shown in Figure 1, our YouTubePD benchmark consists of multiple modalities, including facial expression videos, facial landmarks, and audios, across 200+ subject videos containing dense and diverse annotations from a clinical expert. Our annotations encompass both video-level and frame-specific information with informative region and textual descriptions. In addition, our dataset spans a number of years for each subject, providing natural PD progression information.
2. We define three challenging tasks for our YouTubePD benchmark: Facial-expression-based PD classification, multimodal PD classification, and PD progression synthesis.
3. We present a comprehensive set of baselines for the three proposed tasks. In contrast to prior deep learning methodologies presented for PD analysis, our baselines leverage more recent, state-of-the-art techniques for enhanced video and multimodal understanding. We demonstrate that these models developed on our benchmark *transfer well to real-world clinical datasets* and exhibit strong performance on the PD classification task.

## 2   Related Work

In this section, we discuss prior work and studies on facial expressivity and hypomimia in PD, as well as previous vision-based frameworks used to address PD classification.

| Dataset | # Videos | # Images | # PD Subjects | # Healthy Controls | Open Access | Modality |
|---|---|---|---|---|---|---|
| Abrami et al. [3] | 107 | – | 68 | – | ✗ | Video |
| Bandini et al. [5] | 306 | – | 17 | 17 | ✗ | Video |
| Grammatikopoulou et al. [19] | – | 6236 | 221 | 1071 | ✗ | Image |
| Jin et al. [25] | 176 | – | 33 | 31 | ✗ | Video |
| Novotny et al. [33] | 166 | – | 91 | 75 | ✗ | Video |
| Su et al. [41] | 172 | – | 47 | 39 | ✗ | Video |
| FacePark-GITA [18] | 270 | – | 30 | 24 | ✗ | Video |
| YouTubePD (Ours) | 283 | – | 16 | 89 | ✓ | Video, Audio, Landmark |

Table 1: Comparison of statistics between datasets used in prior work and our benchmark. Our YouTubePD is the first *open-access* and *multimodal* benchmark for PD analysis.

**Facial expressivity in PD.** The connection between PD and facial expressivity has been thoroughly studied in prior work across a variety of experimental scenarios and controlled variables [9, 37, 40, 44, 47]. Such work explores the role of facial expressivity (both posed and voluntary expressions) and emotion recognition in patients in relation to their diagnosis of PD, primarily based on the UPDRS-III scale [16]. Follow-up work [33, 38, 44, 45, 47, 48] expands upon these seminal studies on facial expressivity in PD, further exploring its relationships with the subjective emotional experience, the correlation with PD severity, and de-novo conditions.

**Video-based PD assessment.** Video-based PD assessment techniques are typically categorized into geometric approaches [5, 19, 25, 33, 41] and appearance-based approaches [3, 17]. The former type employs low-dimensional, geometric features extracted from facial landmarks and action units (AUs) [5, 17, 19, 33, 41]. The latter type uses the raw visual information contained within the videos. Some of these approaches apply convolutional neural networks [3, 17], support vector machines [5], and long short-term memory (LSTM) models [25].

**PD benchmarks.** Previous studies commonly use *their own (private or semi-private)* benchmarks to evaluate the performance of their methods, lacking a shared benchmark for comparison. Table 1 provides a summary of these datasets, which often exhibit variations in size and distribution (e.g., mean age, gender, and disease progression). Consequently, comparing the performance of different methods becomes challenging, further hindering advancements in PD classification and analysis. This is a key motivation for our work in establishing a public and common benchmark.

## 3   Dataset

Our objective is to identify not only the presence but also the severity of Parkinson's Disease (PD). With this in mind, we introduce YouTubePD, which stands as the *first publicly available* benchmark designed for PD analysis that (i) utilizes *multimodal* information including in-the-wild videos, audios, and facial landmarks, and (ii) enables the exploration of *various task types* including unimodal and multimodal PD classification, as well as PD progression synthesis. The comparison between our benchmark and datasets in various previous studies is summarized in Table 1. To further support comprehensive classification approaches, we offer video-level and region-level clinician annotations pertaining to PD diagnosis. In the interest of open access and wider research contributions, the basis of this benchmark is sourced from YouTube interview videos featuring public figures who are openly sharing their experiences with PD. Our work is among the first endeavors advocating for PD analysis using multimodal information. This approach aligns with the diagnostic techniques employed by human clinicians, who utilize a wide range of cues in order to diagnose PD. This convergence of multimodal information in our model mimics the human approach, thereby enriching the accuracy and depth of PD analysis. Our dataset collection and annotation pipeline is outlined in Figure 2.

**Videos.** We manually curate a list of 16 public figures with a confirmed and open PD diagnosis and collect multiple videos (spanning multiple years) of the individual before and after their diagnosis. We collect a total of 283 videos from YouTube. Each video is roughly 10 seconds long of an individual public figure speaking in an interview setting. Our corpus can be divided into three subgroups: 65 videos of public figures after their diagnosis of PD was made public, 68 videos of the same public figures several years before their diagnosis, and 150 videos of a broader healthy control group of public figures without PD. We collect and store both video and audio data. The videos are preprocessed by cropping and resizing with a facial keypoint detection model [29], so that clips are centered on the subject's face.

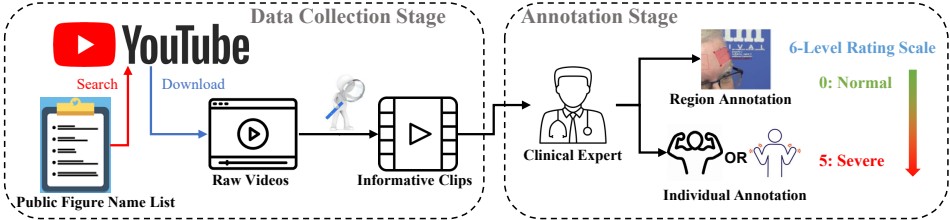

Figure 2: Our dataset collection and annotation pipeline. First, we compile a list of public figures who have publicly confirmed their PD diagnosis. We then source their videos from YouTube. From these videos, we handpick clips that are informative for PD detection. A clinical expert then reviews these clips, providing both video-level and region-level annotations, detailing the severity of their PD, and highlighting specific symptoms of the condition.

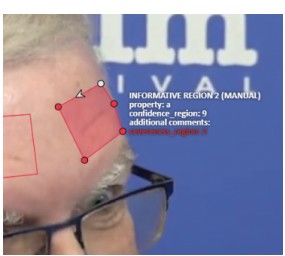

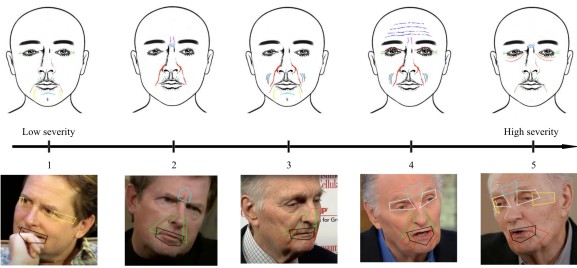

(a) An example annotation of a video from our dataset.

(b) Examples of annotated frames with corresponding bounding boxes in increasing severity.

Figure 3: Representative examples for our video annotations and important facial regions.

**Annotations.** Our annotations (Figure 3) are generated by a clinician. Each video has both (i) an overall severity label with 6 levels, with 0 denoting the absence of PD and 5 indicating a severe form of PD; and (ii) a confidence level label ranging from 1 to 10. A label of 1 indicates that the clinician was not confident in their assessment, while 10 indicates absolute confidence. For selected frames in each video, facial regions particularly informative for PD are also annotated. We define a set of 14 important facial regions for PD analysis, derived as a combination of facial creases [20] and other areas of facial movement observed in PD hypomimia. Each region is mapped to a distinct region index that describes the facial region used for the diagnosis and 10 distinct symptom indices (e.g., moving, apart, increasing) that describe the anomaly. In these frames, (i) a polygon bounding box is drawn around the region, (ii) a text caption is used to summarize the region and anomaly, and similar to the video-level annotations, (iii) a severity label and (iv) a confidence label are provided for each region. These annotations and indices are summarized in Section B of the Appendix.

**Audios.** We provide audios as an additional modality. Each video clip has its associated audio sharing the same label. The audio clips are preprocessed to account for cases where multiple speakers are present or significant background noise or music overpowers the intended speaker. We utilize source separation [42] to clean any segments that have either of these two violations, isolating the desired subject's audios from other speakers and background noise. Though the audio modality has a distinct set of UPDRS standards [16], we do not re-annotate the ground truth label for the audios but use the holistic severity and confidence annotation based on the videos instead. This aims to ensure the *label consistency* between the audios and other modalities to facilitate multimodal PD classification.

**Landmarks.** Our dataset incorporates facial landmarks extracted from video frames to leverage facial motion. Using the landmark detection model from Face++ [29], we extract 106 2D point landmarks from each frame. The landmarks can be grouped into 7 parts: contour, left/right eyebrow, left/right eye, nose, and mouth.

## 4 Tasks

We propose three tasks on YouTubePD: facial-expression-based PD classification, multimodal PD classification with facial expressions, audios, and facial landmarks, and PD progression synthesis. We describe our tasks, as well as the dataset splits and evaluation metrics for each task.

## 4.1 Facial-Expression-Based PD Classification

**Task description.** Following prior work [17, 18] and clinical findings [25], we focus on facial expression as the primary modality for diagnosing PD. In this task, we aim to analyze the videos of facial expressions in our dataset and obtain a classification result for either a binary classification of PD or healthy, or a multiclass classification of PD severity on healthy or 5 severity levels.

**Dataset split.** Due to the imbalanced distribution between PD-positive and PD-negative videos, we use a relatively small training set and a relatively large evaluation set. For the training set, we randomly sample 36 PD-positive and 36 PD-negative videos. We use the remaining 211 videos for the evaluation set. Due to the small number of annotated PD-positive samples, we use K-fold cross-validation on the training set for hyperparameter tuning.

**Evaluation metrics.** The PD classification task on our benchmark is particularly challenging, due to its strong imbalance and limited number of positive samples. This mirrors the medical application setting of PD and presents a challenge: a model whose errors consist of mostly false negatives might still achieve a high accuracy due to the small number of positive samples, but may misdiagnose a patient with PD. This is a more harmful error than a false positive. Therefore, the conventional classification accuracy may not be the best choice for our setting. We mainly focus on additional metrics, in particular, F1 and AUROC. F1-score calculates the harmonic mean of precision and recall. For multiclass classification, we report the weighted F1-score computed with a one-vs-rest (ovr) approach. AUROC computes the area under the receiver operating characteristic curve for a set of decision thresholds and reports a summarized metric through some averaging policy (e.g., macro).

Meanwhile, we classify the severity of PD, which is an *ordinal* attribute. For example, classifying severity 4 as 3 is more accurate than classifying it as 2. Therefore we also provide a Mean Squared Error (MSE) to account for this issue. To summarize, we recommend *prioritizing F1-score and AUROC, while also paying attention to the MSE error*.

## 4.2 Multimodal PD Classification

In this task, we consider the combination of multimodal information for PD classification. The multimodal inputs in our study include facial expression videos, facial landmarks, and audios. The task aims to obtain a binary or multiclass severity prediction from the collaboration of all available modalities as input. We follow the identical data split and evaluation metric as the facial-expression-based PD classification task. We also directly use the holistic label from the facial expression video as the ground truth label for each modality derived from that video.

## 4.3 PD Progression Synthesis

**Task description.** We further propose a task of PD progression synthesis, which aims to simulate the symptoms of PD in facial expressions given images of healthy individuals. Previous work on image translation primarily focuses on texture alterations derived from a reference image or content modifications from a pretrained conditional model [11, 15, 30, 35, 49, 50]. Our PD progression synthesis task, however, introduces a more challenging aspect by incorporating more nuanced PD-specific information for the transfer. Our task naturally makes use of the PD progression information provided in our dataset, which contains sets of images depicting individuals in both healthy and PD states across an extended time frame. Note that these image pairs from healthy to PD states are not strictly aligned, which further increases the task difficulty. Complementary to the discriminative tasks in Sections 4.1 and 4.2, our synthesis task provides a comprehensive overview of the progression and manifestation of PD.

**Dataset split.** We divide the data by individual subjects, allocating 11 for training and 5 for evaluation. Our experiment validates that this split is sufficient for training and produces reliable evaluation results.

**Evaluation metrics.** To establish baselines for our synthesis or "style" transfer task, we employ a variety of metrics that assess the quality of synthesized facial images. Our evaluation begins with the Fréchet Inception Distance (FID) score computed between the source and generated images, serving as an indicator of visual fidelity and consistency. Subsequently, we devise a facial content change evaluation rooted in the paired healthy and PD states of the same individual, called *direction score*. Specifically, we use a pretrained VGGFace [36] model $\phi$ to extract the instance-level feature from each image. For each healthy frame $N_i$, we first identify its corresponding frame $P_i$ from the same

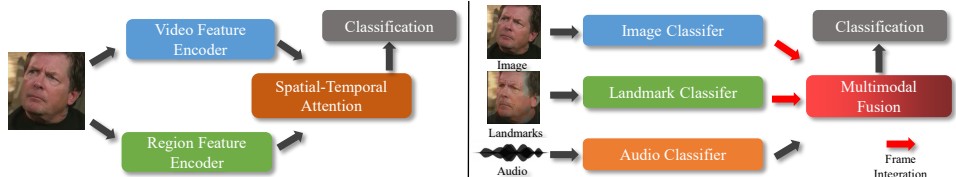

Figure 4: Our baseline methods for the facial-expression-based PD classification (**left**) and multimodal PD classification (**right**) tasks.

individual's PD frame with the highest feature similarity. Then, given the generated image $N_i'$, the direction score is calculated as

$$D = \frac{1}{M} \sum_{i=1}^{M} \text{cosine}(\phi(N_i') - \phi(N_i), \phi(P_i) - \phi(N_i)), \tag{1}$$

where $\text{cosine}$ denotes the cosine similarity and $M$ is the number of frames. Finally, we assess a CLS score based on classification accuracy using a pretrained PD classification model in Section 4.1.

## 5 Methods

In this section, we propose baseline methods for each of our three tasks. For facial-expression-based classification, we propose an attention-based architecture that leverages both region and video annotations for stronger and more interpretable performance. For multimodal PD classification, we establish a simple fusion pipeline to exploit different modalities and improve performance. Finally, for PD progression synthesis, we make comparisons of various image translation models. The experimental results reveal limitations of different methods and the challenging nature of the tasks. We include illustrations of our baseline methods for the first two tasks in Figure 4.

### 5.1 Facial-Expression-Based PD Classification

We develop a simple but effective baseline method for this task. Apart from leveraging the most suitable pretrained representation, we integrate the region-level information explicitly alongside the holistic video information. Moreover, we apply an attention-based classifier and use a novel multiclass hierarchy-guided loss to train our baseline model. Our pipeline is in Figure 4.

**PD representation learning.** Due to the lack of annotated PD data, we find that it is beneficial to transfer representations learned from related domains to PD classification. To this end, we use a ResNet50 [22] pretrained on VGGFace [36], a face dataset with a wide range of subjects, as our frozen feature backbone. Pretraining on the base domain helps the model learn a wide range of identities across variations in pose, environment, and demographics. We find that for our task, pretraining on general facial recognition outperforms backbones trained on video action recognition as well as facial expression recognition, contrary to [18].

**PD informative regions.** Our PD informative regions are based on areas of the face which experts typically use to diagnose patients. Our quantitative results show that the use of region features instead of entire frames not only improves performance, but also reduces the tendency to learn spurious features. Correspondingly, we use a pretrained facial landmark classification module and RoIAlign [21] to extract a feature map for each of the 14 PD informative regions from the feature map produced by the backbone from each video.

**Attention with video-level conditioning.** Following feature extraction, we process the feature maps with a learnable positional embedding and a linear projection of the global feature map to the same dimension as an individual region feature map. Next, we pass our region features through two multi-headed attention blocks. Following TimeSFormer [7], we use divided spatial-temporal attention, which we apply on the region feature maps instead of raw pixels. This consists of an attention block that applies self-attention over the individual regions in each frame (*spatial*), followed by an attention block that applies self-attention over successive frames (*temporal*). We include ablation studies on the use of region annotations, region information, and attention to verify each component of our method.

**Multitask hierarchy-guided loss.** In addition to region features, we leverage region annotations to further improve performance. After attention, we extract the class embedding corresponding to each

| Model | Top-1 Acc ↑ | F1 ↑ | AUROC ↑ | Recall (Binary) ↑ | MSE (Multiclass) ↓ |
|---|---|---|---|---|---|
| VGGFace [22] | 83.56(±0.84)/78.20(±3.13) | 0.56(±0.01)/0.23(±0.02) | 0.86(±0.01)/0.68(±0.01) | 0.79(±0.01) | 2.29(±0.77) |
| Ours | 88.00(±1.88)/77.03(±3.36) | 0.59(±0.02)/0.25(±0.01) | 0.92(±0.01)/0.74(±0.02) | 0.85(±0.05) | 2.25(±0.45) |

Table 2: Binary/multiclass classification results on YouTubePD with facial expression. We observe superior performance on F1, AUROC, and MSE with our method.

| Modality | Top-1 Acc ↑ | F1 ↑ | AUROC ↑ | Recall (Binary) ↑ | MSE (Multiclass) ↓ |
|---|---|---|---|---|---|
| VGGFace [22] | 83.56(±0.84)/78.20(±3.13) | 0.56(±0.01)/0.23(±0.02) | 0.86(±0.01)/0.68(±0.01) | 0.79(±0.01) | 2.29(±0.77) |
| Landmark (Ours) | 56.37(±1.72)/51.69(±1.94) | 0.32(±0.02)/0.26(±0.02) | 0.72(±0.02)/0.68(±0.01) | 0.81(±0.05) | 4.87(±0.28) |
| Audio (Ours) | 54.84(±1.63)/47.79(±1.66) | 0.27(±0.02)/0.16(±0.01) | 0.66(±0.01)/0.50(±0.02) | 0.70(±0.04) | 6.26(±0.39) |
| Multimodal (Ours) | 70.61(±2.13)/82.75(±2.85) | 0.61(±0.02)/0.28(±0.02) | 0.87(±0.02)/0.80(±0.03) | 0.83(±0.04) | 1.40(±0.25) |

Table 3: Binary/multiclass classification results on YouTubePD in the multimodal setting. Multimodal fusion further improves the performance over unimodal baselines, even when additional modalities have lower performance than the primary modality of facial expression.

region. We add an additional multi-layer perception (MLP) head for each of the 14 regions and add the corresponding cross-entropy loss to the overall loss objective. We convert the region annotations to binary annotations, where 1 indicates that the region is indicative of PD and 0 indicates that the region is normal. We also extract a class embedding corresponding to the entire video, which is passed through a final MLP head to obtain the video-level class distribution. This is trained with a hierarchy-guided loss function demonstrated by the weighted cross-entropy loss, which we add to the region-level loss,

$$\text{Loss} = \lambda \, l_{\text{bin}}(\theta(x), \, y_{\text{vid}}) + (1 - \lambda) \, l_{\text{multi}}(\theta(x), \, y_{\text{vid}}) + \frac{1}{|R|} \sum_{r \in R} l_{\text{reg}}(\theta(x_r), \, y_{\text{reg}}), \quad (2)$$

where $x$ is an input image, $\theta$ is the model, $y_{\text{vid}}$ is the video-level annotation, $y_{\text{reg}}$ is the region-level annotation, $l_{\text{bin}}$ and $l_{\text{multi}}$ are the cross-entropy losses on binary and multiclass labels respectively, $l_{\text{reg}}$ is the cross-entropy loss on region labels, $R$ is the set of regions, and $\lambda$ is a trade-off hyperparameters. We use both the original multiclass annotations and binary annotations for the video-level loss. We freeze the first three stages of the pretrained ResNet50 backbone and train the additional layers, as well as our attention blocks and MLP heads, end-to-end with this loss.

## 5.2  Multimodal PD Classification

We provide a baseline multimodal fusion method for PD classification. As shown in Figure 4, we first obtain predictions from each modality, and then use a logit fusion step to fuse them together considering their respective confidence.

**Single modal prediction.** For the facial expression video modality, we utilize the same encoder as in the facial-expression-based PD classification (Section 5.1), which yields a 2,048-dimensional feature representation. We obtain the final prediction with a linear classifier. For the landmark modality, we select 10 frames from the video, extract their facial landmarks, and classify them using an MLP. The features from all selected frames are concatenated for classification. For the audio modality, we extract the features using a pretrained masked auto-encoder applied on the spectrogram representations of the audios [32]. After averaging across all frames to get a 768-dimensional video-level feature representation, an MLP is trained to obtain the video-level logits.

**Fusion strategy.** We employ a simple but effective fusion strategy. For the video and landmark modalities, rather than using concatenation for the classification, we predict each frame independently. Then we sum the predictions of 10 frames to obtain two logits representing the video modality and landmark modality. After that, we apply $\mathcal{L}_1$ normalization to the logits of each modality. Thus, we treat the logits for different modalities equally. Then we directly sum them up and determine the final prediction based on the max fused logits. During training, the same multitask hierarchy-guided loss is adopted. For the facial expression and audio modules, all the pretrained weights are frozen.

## 5.3  PD Progression Synthesis

We investigate a wide spectrum of image translation methods, including generation-based generative adversarial networks (GANs), translation-based GANs, and recently emerging diffusion models. Specifically, for generation-based GANs, we adopt [11, 26, 34]. We use their pretrained weight on FFHQ [27] and finetune it on our positive training data. Then, we project the PD-negative face to the

| Metric | Generation-Based GAN | | | Translation-Based GAN | | | Diffusion Model |
|---|---|---|---|---|---|---|---|
| | ADA [26] | Few-Shot [34] | JoJoGAN [11] | HRFAE [49] | CycleGAN [50] | CUT [35] | SD [39] |
| FID ↓ | 87.08 | 94.60 | 129.61 | 37.65 | 73.82 | 46.63 | 72.12 |
| CLS ↑ | 34.29 | 26.67 | 37.74 | 11.32 | 26.42 | 11.32 | 66.98 |
| Direction ↑ | 0.2874 | 0.2912 | 0.3219 | 0.2021 | 0.2832 | 0.3237 | 0.0491 |

Table 4: Quantitative comparisons on PD progression synthesis. Different generative methods exhibit distinct behaviors under three metrics.

| Model | Top-1 Acc ↑ | F1 ↑ | AUROC ↑ |
|---|---|---|---|
| VGGFace [22] | 79.4 | 0.24 | 0.66 |
| Ours | 76.4 | 0.25 | 0.71 |

Table 5: We observe similar multiclass classification performance, when conducting a leave-one-subject-out analysis over three patients.

learned image space using the GAN model. For GANs designed for image translation, we either use their official pretrained model [49] or directly train from scratch [35, 50]. For the diffusion models, we finetune Stable Diffusion [39] with LoRA [24] and use SDEdit [30] for image translation.

## 6 Experiments

In this section, we present the results of our baselines and proposed methods. We find that modern computer vision methods can be applied on YouTubePD and transfer well to real clinical applications.

**Main results: Facial-expression-based PD classification.** We report the average results and standard deviations over 5 different runs. The results are summarized in Table 2. For the baseline, we use a ResNet50 pretrained with VGGFace and finetuned on our dataset with linear probing. This obtains relatively strong performance, suggesting the efficacy of facial representations for PD diagnosis. In both the binary and multiclass settings, our proposed attention-based model outperforms the pretrained VGGFace baseline on most metrics.

**Main results: Multimodal PD classification.** We additionally provide baseline results for the other modalities as described in Section 5. From Table 3, we note that either modality alone ends up being considerably weaker than our primary modality of facial expression, and the fusion result achieves better performance in both the multiclass and binary settings. Our results primarily serve to demonstrate the potential for achieving better performance from the additional modalities in tandem with facial expressions, despite their weaker performance when used in isolation.

**Main results: PD progression synthesis.** The synthesis results of various baselines are shown in Table 4 and Figure 5. FID represents the consistency of the image translation. The CLS score and direction score focus on the semantic change in different aspects. Generally speaking, generation-based GANs often obtain balanced results on CLS and direction metrics. However, they fail to achieve low FID, since the image needs alignments to pretrained data. Translation-based methods can often get high consistency, but are easy to focus on other modifications rather than PD progression. The diffusion models achieve a significantly higher CLS score; however, they tend to overfit the training data and create faces that are similar to training images but do not retain the facial features of the test image. Thus, they will largely change the faces in test images, resulting in a low direction score. Our extensive comparisons reveal the substantial limitations of state-of-the-art generative models in extracting very fine-grained and subtle information for image translation, further highlighting the importance of our benchmark.

**Analysis: Improvements are robust to different dataset splits.** In Table 5, we present the results when we train the facial expression model on a modified training and test split, where we ensure there is a PD-positive patient only represented in the test split. We report the average multiclass results of three such leave-one-subject-out splits. In these new splits, we still observe that our method outperforms the baseline under the F1 and AUROC metrics. In the leave-one-subject-out setting, the AUROC of our method improves by 0.05 and F1 improves by 0.01. This improvement is similar to our original setting (AUROC by 0.06 and F1 by 0.02). Please note that similar to the main results in Table 2, the Top-1 accuracy is not indicative of model performance, because of the presence of a large amount of negative samples.

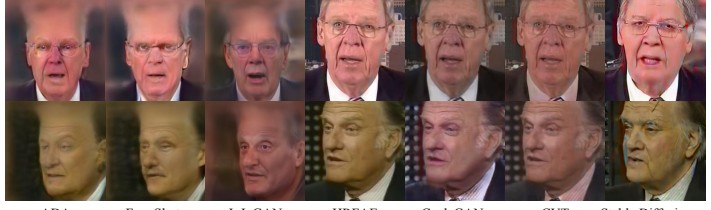

| | | | | | | | | |
|Healthy|ADA|Few-Shot|JoJoGAN|HRFAE|CycleGAN|CUT|Stable Diffusion|PD Positive|

Figure 5: Qualitative comparisons on PD progression synthesis. The diffusion model yields superior results; however, it may inadvertently introduce inaccurate alterations to the face.

| Model | Top-1 Acc ↑ | F1 ↑ | AUROC ↑ |
|---|---|---|---|
| VGGFace [22] | 69.31/71.29 | 0.42/0.45 | 0.79/0.81 |
| Ours | 68.31/77.22 | 0.52/0.61 | 0.92/0.96 |

Table 6: Binary classification results on real-world patients in the *speech test* set and the *facial activation* set [23]. Our method consistently outperforms baselines on real-world PD data.

| Model | Binary Acc ↑ | Multiclass Acc ↑ |
|---|---|---|
| I3D [10] | 70.60 | 45.64 |
| C2D [46] | 69.05 | 40.96 |
| X3D-XS [13] | 74.17 | 42.77 |
| SlowFast-8x8 [14] | 63.81 | 44.52 |
| VGGFace | 83.56 | 78.20 |

Table 7: Ablation study on action recognition pre-training. We observe significantly lower performance than using our facial recognition pretraining.

| Model | Top-1 Acc ↑ | F1 ↑ | AUROC ↑ |
|---|---|---|---|
| AffectNet-8 [31] | 76.00 | 0.45 | 0.86 |
| VGGFace | 83.56 | 0.56 | 0.86 |

Table 8: Ablation study on facial expression classification pretraining, in terms of binary classification with the AffectNet-8 backbone. We observe a decrease in performance compared with when using our facial recognition pretraining.

| Model | Top-1 Acc ↑ | F1 ↑ | AUROC ↑ |
|---|---|---|---|
| Video / Temporal Attention | 77.14 | 0.49 | 0.91 |
| Region / Spatial Attention | 82.86 | 0.56 | 0.90 |
| Ours | 88.00 | 0.59 | 0.92 |

Table 9: Ablation study on binary classification results using various attention schemes. Our spatial-temporal attention performs the best.

**Generalizability: From YouTube to real patients.** From a healthcare perspective, a crucial and natural question arises: *Do the models developed in our YouTubePD benchmark generalize to real-world PD diagnosis?* To answer this question, we train our model on YouTubePD and test it on a private dataset of clinical PD patients [23]. This dataset consists of two sets of videos, a speech test of 18 PD-positive and 83 PD-negative patients, and a facial activation test of 19 PD-positive and 85 PD-negative patients. From Table 6, our proposed method has higher F1 and AUROC scores than the baseline on this dataset. In addition, the t-SNE [43] visualizations in Figure 11 of the Appendix show that the PD-positive and PD-negative samples are better separated for our approach. These experimental results validate the efficacy of our benchmark for real-world applications.

**Ablations.** We conduct ablation studies to investigate the design choices and components of our proposed facial-expression-based PD classification method. These studies specifically focus on understanding the benefits of different types of pretraining and attention for PD classification.

*Action recognition pretraining.* We investigate the transferability of generic video representations to PD classification. We finetune various action recognition architectures pretrained with Kinetics400 [10] on YouTubePD with linear probing. From Table 7, we observe much lower performance compared with generic facial representations from VGGFace [36], despite using a simpler architecture.

*Facial expression classification pretraining.* We also investigate the use of a backbone pretrained for facial expression classification, a more fine-grained task than facial recognition. We use a ResNet50 pretrained on AffectNet-8 [31] as our backone, and finetune it on YouTubePD with linear probing. From Table 8, we observe lower performance with the AffectNet-8 backbone, suggesting that generic facial representations transfer more easily to PD classification.

*Attention types.* We investigate the use of attention over regions and attention over frames. This is combined in our proposed facial-expression-based method. From Table 9, we observe similar overall performance for either temporal or spatial attention and improved performance when combined.

# 7 Discussion

**Clinical impact.** The long-term goal of our work is to develop a Parkinson's Disease (PD) early screening tool to aid primary clinicians in the earlier recognition of persons exhibiting physical signs

that may indicate an evolving Parkinson's syndrome. Those persons may then undergo additional neurological evaluation as clinically indicated. This will help patients to be diagnosed and treated sooner. Many patients with Parkinson's, in retrospect, exhibited subtle signs of Parkinson's for months or even years. The signs were not recognized by the patient, their family, or primary clinicians and the signs instead were attributed to aging, and diagnosis was delayed. We hope that our effort could eventually inspire such a PD early screening tool to detect PD symptoms at an early stage, and the patients could get timely treatments.

**Annotation reliability.** In the construction of our dataset, we made use of annotation information beyond what a single annotator provided. Since we did not construct the dataset completely from scratch but repurposed existing YouTube videos, we explicitly utilized the topic of these YouTube videos and their meta-information. Therefore, we precisely know which videos are PD-positive and which ones are negative, and there will very unlikely be mistakes in the annotation of video-level PD labels. In addition, the longitudinal meta-information associated with the YouTube videos provides useful prior knowledge about the PD severity. Empirically, models trained on our annotated dataset generalize to real-world clinical PD data as shown in Table 6. Our method achieves an F1 score that is 10% and 16% higher than the baseline on two different sets of real patient data, respectively. These results reflect that our annotation should be reliable.

**Limitations and future work.** The dataset presented in our benchmark possesses its own caveats, particularly in relation to its small size and limited diversity due to the constraint of publicly available videos. While our benchmark serves as a strong first step, there is still a need for further comprehensive datasets and benchmarks to establish thorough and holistic evaluation of models developed for PD analysis. In addition, an important question remains regarding the generalizability of models developed on our benchmark to critical, real-world medical applications across various settings. Further exploration and analysis of this aspect are necessary to advance towards the actual deployment of these models in clinical scenarios.

Beyond providing the dataset, our contribution includes the establishment of a crucial *protocol* for collecting a PD dataset. This ensures that our dataset can be readily expanded in the future with more public figures or individuals from additional geographical regions and across different social media platforms other than YouTube. We hope that our work will inspire efforts to create larger PD benchmarks with more annotators. We discuss our limitations and future work in more detail in Section E of the Appendix.

**Ethics.** Our research involves the analysis of videos featuring public figures with PD. The videos, derived from YouTube, were publicly shared by figures who had willingly discussed their PD condition. To ensure ethical considerations, we sought explicit consent from these figures. In addressing concerns of data privacy, the research protocol was reviewed and approved by the Institutional Review Board (IRB) at University of Illinois Urbana-Champaign (IRB approval number: 24426). While we acknowledge the potential biases and limitations in solely relying on facial or audio information for PD diagnosis, *our main objective is to inspire tools for early detection using easily accessible media like webcams*. This initiative does not negate the need for comprehensive medical diagnostics. We stress that our efforts aim to further the understanding of PD and its facial expression impacts. The collected data are shared with researchers who possess ethical training and commit to adhering to high standards. All data are hosted on a dedicated and secure platform, and the code is made available on GitHub. No conflicts of interest exist among the study's contributors. More discussion on the ethical aspect of YouTubePD is included in Section F of the Appendix.

## 8   Conclusion

We introduce YouTubePD, the first publicly available multimodal benchmark for Parkinson's Disease (PD) analysis. Within this framework, we propose several important discriminative and generative tasks along with corresponding baselines, showcasing how a range of modern machine learning and computer vision techniques is leveraged and extended to advance AI-assisted systems for automated PD diagnosis. Furthermore, we demonstrate that models trained on YouTubePD perform well on clinical data, indicating potential for real-world medical applications. Notably, the methodology and protocol developed in constructing YouTubePD can be adapted to address other healthcare problems, where a public, standardized benchmark is missing. We encourage further exploration in this direction and hope that our benchmark will facilitate a more seamless transfer of advancements in machine learning and computer vision to impactful medical research and clinical applications.

## Acknowledgements

This work was supported in part by the Jump ARCHES endowment through the Health Care Engineering Systems Center, the National Center for Supercomputing Applications (NCSA) at the University of Illinois at Urbana-Champaign through the NCSA Fellows program, NSF Grant 2106825, and NIFA Award 2020-67021-32799. We thank Minh Do and the team for providing real-world Parkinson's patient data.

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

## Appendix

In this appendix, we include (1) discussion regarding our YouTubePD benchmark detail, release, and licensing, (2) additional analysis of the dataset, (3) additional illustration and details about our methods, (4) additional experimental results and analysis, (5) more discussion on limitations and future work, and (6) potential negative societal impact.

## A    Benchmark Detail, Release, and Licensing

We provide our code and benchmark at <https://uiuc-yuxiong-lab.github.io/YouTubePD>. We release our benchmark under the CC0 license. Here, we describe how we publicly release our benchmark:

1. We include all the code used in the process of converting publicly available YouTube videos into our benchmark.

2. We include all our annotations and extracted landmarks. Note that offensive content is not included in our dataset, as all the sources are publicly available interviews of public figures speaking openly about their experiences with Parkinson's Disease.

Despite that all the videos used in our benchmark are publicly available YouTube videos, we are also actively taking steps to approach the public figures involved to respect their autonomy and privacy. This ensures that we uphold the highest standards of ethical data usage. We strive to balance open access and reproducibility with respect for privacy, all while providing a resource that could significantly advance the analysis and understanding of Parkinson's Disease (PD).

## B    Dataset: Additional Analysis

### B.1    Dataset Statistics

| Region/Severity | 0 | 1 | 2 | 3 | 4 | 5 |
|---|---|---|---|---|---|---|
| Overall Video | 187 | 15 | 8 | 10 | 17 | 7 |
| Forehead | 187 | 12 | 3 | 4 | 16 | 18 |
| Left nasolabial fold | 187 | 9 | 6 | 10 | 20 | 10 |
| Right nasolabial fold | 187 | 11 | 7 | 10 | 21 | 8 |
| Right lip crease | 187 | 1 | 0 | 3 | 7 | 3 |
| Left lip crease | 187 | 1 | 0 | 2 | 10 | 3 |
| Left outer eye | 187 | 4 | 2 | 1 | 6 | 17 |
| Right outer eye | 187 | 2 | 2 | 2 | 5 | 16 |
| Between eyebrows | 187 | 5 | 1 | 3 | 5 | 3 |
| Right above between eyebrows | 187 | 3 | 3 | 4 | 12 | 5 |
| Right eye | 187 | 15 | 4 | 3 | 8 | 6 |
| Left eye | 187 | 16 | 4 | 3 | 8 | 6 |
| Mouth | 187 | 1 | 0 | 1 | 5 | 0 |
| Right eyebrow | 187 | 6 | 5 | 8 | 11 | 12 |
| Left eyebrow | 187 | 6 | 4 | 7 | 10 | 12 |
| Total Annotations | 2808 | 107 | 49 | 71 | 161 | 126 |

Table 10: Distribution of severity labels in YouTubePD for the overall video-level analysis and for all 14 facial regions, with 0 denoting the absence of PD and 5 indicating a severe form of PD.

| Country | Count | Race | Count | Gender | Count |
|---|---|---|---|---|---|
| United States | 12 | White/Caucasian | 13 | Male | 15 |
| United Kingdom | 3 | Black/African descent | 3 | Female | 1 |
| Canada | 1 | | | | |

Table 11: Country/race/gender statistics of PD-positive public figures in YouTubePD.

| Country | Count | Race | Count | Gender | Count |
|---|---|---|---|---|---|
| South Africa | 1 | South African+Swiss-German | 1 | Male | 68 |
| United States | 52 | Black/African Descent+Filipino | 1 | Female | 23 |
| United Kingdom | 12 | Black/African Descent | 11 | | |
| Israel | 3 | White/Caucasian | 63 | | |
| Russia/Canada | 1 | Indian | 1 | | |
| Sweden | 2 | Latinx | 4 | | |
| Kenya/Mexico | 2 | Asian | 9 | | |
| Brazil | 2 | Black/African Descent+Samoan | 1 | | |
| Serbia | 1 | | | | |
| South Korea | 2 | | | | |
| Puerto Rico | 1 | | | | |
| Mexico | 1 | | | | |
| Japan | 1 | | | | |
| Canada | 4 | | | | |
| Denmark | 1 | | | | |
| Russia | 2 | | | | |
| Germany | 1 | | | | |
| France | 2 | | | | |

Table 12: Country/race/gender statistics of healthy control or PD-negative public figures in YouTubePD.

In Table 10, we summarize the severity label distribution in YouTubePD. This includes severity labels for the overall subject in each video and severity labels for each of the 14 important facial regions informative for PD analysis. The number of annotations varies depending on severity levels and regions.

We also summarize the demographic distribution in YouTubePD, split between PD-positive and healthy control (HC), or PD-negative, subjects. Table 11 provides the country, race, and gender statistics of PD-positive subjects in YouTubePD. Similarly, Table 12 provides the country, race, and gender statistics of HC subjects in YouTubePD.

We would like to provide additional details in our annotation process, particularly regarding how we denote the severity of PD. Our annotation strategy utilizes a detailed scale, ranging from 0 to 5, where 0 signifies a healthy individual, and 5 corresponds to severe PD. We do not apply the Unified Parkinson's Disease Rating Scale (UPDRS) [16] for facial expression. This decision is based on the clinician's suggestion, since an accurate UPDRS facial expression rating would require more information (e.g., observing the subject's facial expression pattern at rest or when not talking) than facial expression videos contain. This strategy also allows for a finer classification. In addition, we do not apply UPDRS because facial expression and audio have distinct UPDRS standards. We instead use the holistic severity and confidence annotation based on the video. Doing so ensures the label consistency between the audio data and other modalities, thereby facilitating multimodal PD classification.

In addition, we provide (i) illustrations of our facial landmarks and regions in detail in Figures 6, 7, and 8; (ii) the longitudinal statistics of our PD videos showing their distribution in time in Figure 9.

## B.2 Are Annotated Regions Correlated to PD Severity?

To understand if the annotated regions are informative for PD severity, we investigate the correlation between the 14 annotated facial regions and a patient's annotated PD severity level. We accomplish this by training a linear classifier that takes as input the annotated informative region index and predicts the severity level. More specifically, for each video, the input is a binary vector which maps the clinician-annotated facial region indices, while the output is matched to the annotated severity level of the video. As a control experiment, we instead train on random region-level annotations as input. We find that the linear model trained on the actual region annotation achieves $70\%$ test accuracy, while the model trained on random annotations achieves $15\%$ test accuracy. This validates that the severity level is predictable from the annotated informative regions, but not from random region annotations. Therefore, *the annotated regions and PD severity are correlated*. This is also

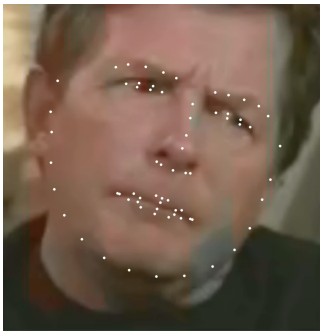

Figure 6: Original landmark extraction.

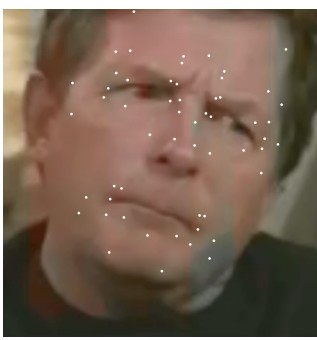

Figure 7: Interpolated landmarks.

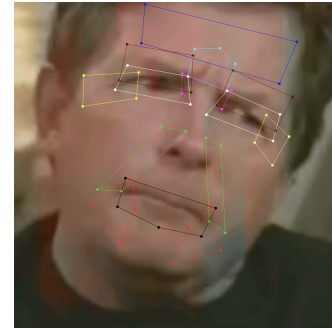

Figure 8: Visualized regions from interpolated landmarks.

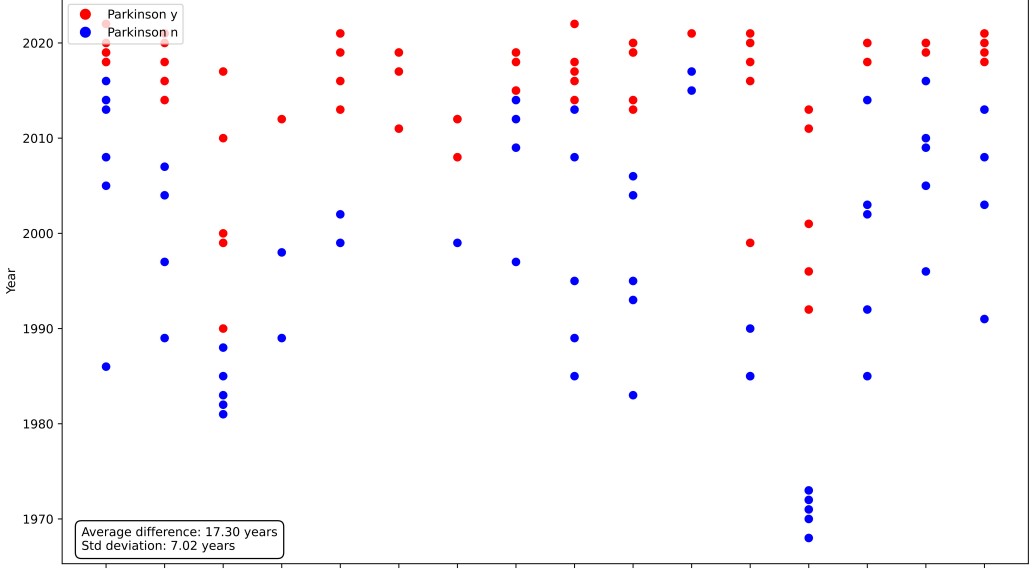

Figure 9: Longitudinal data for the time gap between PD-negative and PD-positive videos. The x-axis represents public figures, while the y-axis represents the time in years. Red dots denote videos with PD-positive labels, and blue dots denote videos with PD-negative labels.

consistent with our approach that leverages region-level information for improved PD classification performance.

Furthermore, we can examine the learned weights and biases of the linear model to understand how the model has learned to classify. We find that in general, higher severity patients have positive annotations on a larger number of informative regions (more symptoms), and vice versa. The model also leverages different facial regions to determine severity at different levels; for example, very severe cases could be easily distinguished via eyes and lips feature. Figure 3 in the main paper visualizes the most informative regions at each severity level. Corresponding regions are indicated by highlighted facial creases in the figure and bounding boxes in the input video frames.

## C   Methods: Additional Illustration and Details

### C.1   Model Architecture Illustration

We provide an additional illustration of our baseline method for the first task in Figure 10. An input video is processed through two branches, one for video features and the other for region features. These features are then aggregated with a spatial-temporal attention classifier to obtain the PD classification result.

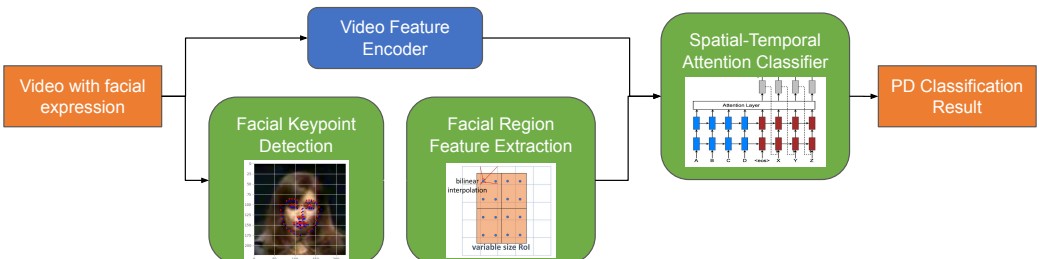

Figure 10: Illustration of our baseline method for the facial expression-based PD classification. An input video is processed through two branches, one for video features and the other for region features. These features are then aggregated with a spatial-temporal attention classifier to obtain the PD classification result.

## C.2 Implementation Details

For the facial-expression-based classification method, we use $\lambda = 0.9$, emphasizing the binary portion of the loss. We use all 14 regions, i.e., $|R| = 14$. We use 8 frames from each video and a batch size of 32. We train our models with the Adam [28] optimizer with a learning rate of 0.001. More details are provided in the code. All experiments corresponding to this task are conducted on a single 16GB NVIDIA V100 GPU.

For the audio baseline, we utilize the entire audio and average the representations to obtain a single 768-dimensional audio-level feature vector. Afterwards, we use a batch size of 64 and feed the input through an MLP with one hidden layer of size 1,024. We train the model using Adam with a learning rate of 0.0003. Similarly, for the landmark baseline, we use 8 frames to maintain consistency with the other modalities, and use the same hyperparameters for Adam. For both modalities, we use $\lambda = 0.5$. All experiments are conducted on a single NVIDIA 3060 GPU.

For the multimodal fusion baseline, we use 8 frames from each video. The batch size is set as 16 and trained for 50 epochs. We train the model using Adam with a learning rate of 0.02. All experiments are conducted on a single NVIDIA 4090 GPU.

For PD progression synthesis, we follow the settings used in the official implementations [11, 26, 34, 35, 39, 49, 50]. In addition, we make modifications to two methods to align with our task setting. Specifically, we replace the age classifier in HRFAE [49] with our trained PD binary classification model. As for JoJoGAN [11], we iteratively sample images during training to utilize all available images. All experiments are conducted on a single NVIDIA 4090 GPU.

## D  Experiments: Additional Results and Analysis

### D.1  Additional Ablations

**Alternative audio representations.** We additionally explore alternative feature representations for audios beyond masked auto-encoders (MAE) [32] presented in the main paper – namely, wave2vec (W2V) 2.0 [4], a deep feature extractor for audios that also uses self-supervised learning, and MEL-frequency cepstral coefficients (MFCC) [6], which have demonstrated reasonable success in general audio processing tasks. We find empirically that MAE features work the best with regards to metrics and consistency, as shown in Table 13. In general, wav2vec features remain competitive with the MAE features though trailing slightly, while MFCC performs considerably worse likely due to its inability to express complex features necessary for the task, given the simplicity of the classification model. Note that the hand-crafted MFCC achieves a relatively high performance on AUROC, since it effectively makes random guess among all classes. The other *learned* features are more biased by the data imbalance between 0 and all other classes. This is due to the fact that AUROC is computed as an

unweighted average of one-vs-all calculations, leading to MFCC to incorrectly appear competitive on that metric.

| Audio Feature | Top-1 Acc ↑ | F1 ↑ | AUROC ↑ | MSE ↓ |
|---|---|---|---|---|
| MAE [32] | 47.79(±1.66) | 0.16(±0.01) | 0.50(±0.02) | 6.26 (±0.39) |
| W2V [4] | 57.90(±7.46) | 0.14(±0.02) | 0.43(±0.04) | 4.50(±0.93) |
| MFCC [6] | 39.42(±9.66) | 0.11(±0.03) | 0.49(±0.06) | 6.48(±1.80) |

Table 13: Ablation study on audio representations for multiclass classification on YouTubePD. 'MAE' denotes masked auto-encoders presented in the main paper; 'W2V' denotes wave2vec 2.0; 'MFCC' denotes MEL-frequency cepstral coefficients. We find that MAE provides the most stable and consistent results – we prioritize F1 and AUROC, as the other metrics are influenced by the data imbalance.

**Multimodal fusion strategies.** We conduct an ablation study on the multimodal fusion strategy. Specifically, we explore two different strategies to produce the logits for the facial expression video and facial landmark modalities. One way is frame concatenation (FC) where we concatenate the frame features, and the other is frame voting (FV) where we perform voting to aggregate the result of each frame. Note that FV is used for results in Table 3. For FC, we concatenate frames' features to one vector representing the whole video (either image features from ResNet or landmark coordinates). Then, we train a video-level classifier to obtain the video logits. For FV, we train a frame-level classifier for each frame and average the predictions as video-level logits. For both strategies, the video-level logits from different modalities are averaged to get the final prediction. As shown in Table 14, with FC, the multimodal performance is even lower than the single facial expression modality on F1 and AUROC metrics. By contrast, the FV strategy helps to improve performance.

| Audio Feature | Top-1 Acc ↑ | F1 ↑ | AUROC ↑ | MSE ↓ |
|---|---|---|---|---|
| VGGFace [22] | 78.20(±3.13) | 0.23(±0.02) | 0.68(±0.01) | 2.29(±0.77) |
| Multimodal (FC) | 79.23(±1.94) | 0.21(±0.02) | 0.69(±0.01) | 1.76(±0.18) |
| Multimodal (FV) | 82.75(±2.85) | 0.28(±0.02) | 0.80(±0.03) | 1.40(±0.25) |

Table 14: Ablation study on multimodal fusion strategies for multiclass classification on YouTubePD. 'FC' denotes frame concatenation, and 'FV' denotes frame voting. We find that the frame voting strategy improves the fusion performance, while the frame concatenation strategy even leads to a decrease in performance on F1 and AUROC metrics, compared with the single facial expression modality.

## D.2 t-SNE Visualizations

We provide qualitative results for the performance of our facial-expression-based classification model. We use t-SNE [43] on both YouTubePD and the private clinical dataset [23], shown in Figure 11. Our approach is able to clearly separate the PD-positive and PD-negative classes on both distributions.

# E More Discussion on Limitations and Future Work

**Limitations.** In the main paper, we have briefly discussed the limitations of our work. Here, we provide a more in-depth discussion. First, as our dataset is composed of publicly available YouTube videos of public figures, the subjects and video samples in the dataset may not adequately represent the wide range of individuals affected by PD. The videos primarily capture interview scenarios, which may not effectively showcase the indicative symptoms of subjects, compared with the motor tasks and instructions typically used in medical studies. Furthermore, there is a demographic bias in the dataset subjects, as they are all public figures (predominantly male) with very few available details about their treatment course and disease progression. Meanwhile, we are not aware of the treatment or treatment response experienced by these individuals.

Second, it is necessary to conduct further investigation and analysis of the performance and deployment of models developed using our benchmark in real-world clinical scenarios. In the main paper, we have demonstrated that our method developed on the benchmark exhibits promising results on a clinical dataset. More comprehensive evaluation on additional clinical datasets would validate the broad generalizability of our benchmark and associated models to practical medical applications.

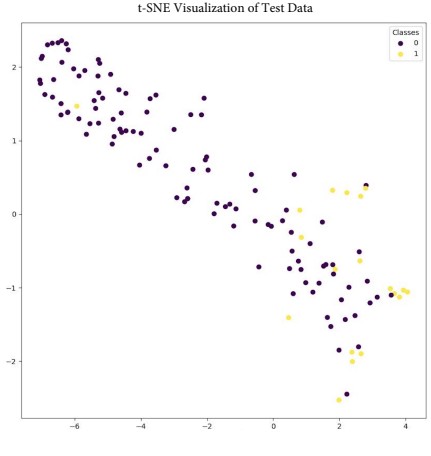

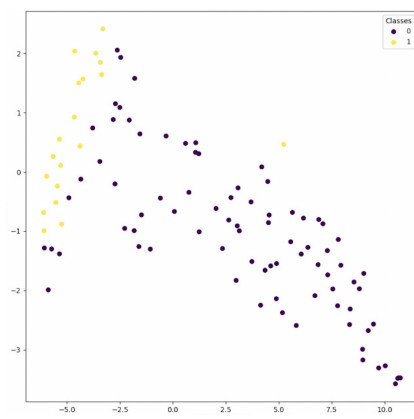

| (a) t-SNE visualization for YouTubePD. | (b) t-SNE visualization for the facial activation set with clinical patients. |

Figure 11: t-SNE visualizations of our learned facial expression representation for healthy (class 0) and PD-positive (class 1) subjects on YouTubePD and the clinical dataset [23]. Intriguingly, we observe a more pronounced separation between the PD-positive and PD-negative classes on clinical data, demonstrating the generalizability of our learned representation from the in-the-wild YouTube videos.

Finally, the size of the dataset is relatively small compared with typical computer vision datasets, due to the inherent challenges involved in collecting PD data. The small dataset size increases the difficulty of developing and training larger models from scratch, often necessitating some form of finetuning to achieve reasonable performance.

**Future work.** These limitations open up a wide scope of future advancements and progress in this field. As mentioned previously, while our benchmark represents a strong first step, further comprehensive datasets and benchmarks are necessary to thoroughly evaluate the performance and generalizability of methodologies prior to their deployment. Moreover, our findings highlight the potential of developing multimodal frameworks that leverage various examination modalities and track complementary symptoms, such as facial expression, speech, posture, and gait for PD classification. Although PD classification has been the primary focus (as in our first two proposed tasks), we note that there are interesting unexplored directions in this realm, particularly in generative tasks like progression synthesis (as in our third proposed task), which can serve as effective augmentation and learning techniques.

## F  Ethics Discussion

### F.1  Personally Identifiable Information and Informed Consent

YouTubePD may include personally identifiable information (PII) or sensitive personally identifiable information. The data we collect from YouTube include facial expressions, PD identity, and audios. However, we would like to highlight that the public figures chose to make their struggles with PD public and discussed their disease and diagnosis in front of cameras. By willingly revealing their identifiable faces and voices, the public figures do not intend to keep their PD information fully private. We believe that the concern regarding a breach of privacy is not a newly raised issue specific to our work, as the possibility of any misuse of these videos already exists.

The central question we posed to ourselves was whether sharing these videos with our research community, along with annotations of facial expressions, would amplify the risk of misuse. We are of the opinion that this action does not escalate the aforementioned risk. To further address this matter, we took the initiative to contact the public figures involved and requested permission. This step was taken proactively, particularly in the event that the public figures had regrets about their previous decision to go public and now wished to make a different choice.

Regarding PII and sensitive PII, we are fully aware of the sensitive nature of the data we are working with. In order to safeguard individuals' privacy, we have sought both guidance from the Institutional Review Board (IRB) Office and legal guidance from the Legal Department at University of Illinois Urbana-Champaign to ensure compliance with regulations. Furthermore, in line with ethical norms, we have made efforts to obtain explicit consent from each public figure featured in the videos. The consent form clearly includes our intention in using these data and how these data are expected to be used. We remove videos of public figures who wish not to be part of our dataset. In addition, we acknowledge the concern about the potential for individuals and their families to feel uncomfortable with the label of "illness." While we respect this sensitivity, we emphasize that our intention is to contribute to a better understanding of PD, its impact on facial expressions, facial landmarks, and audios, and the potential for technological advancements. We approach this research with the utmost respect for the individuals involved and strive to contribute positively to the discourse around the disease.

### F.2  Negative Societal Impact

While our work provides promising advancements in AI-assisted analysis and severity evaluation of PD, we recognize that it may also present potential negative societal impacts that deserve careful consideration.

The first concern pertains to privacy. The videos we use for our work are publicly available, featuring public figures who openly discuss their experience with PD. However, widespread use of similar technology could raise issues of privacy, as individuals may not wish to have their health condition detected or revealed, even inadvertently, through casual video or audio footage. As healthcare professionals and researchers, it is critical to respect patient privacy and consent in all facets of care and study. Second, there is a risk of misuse or over-reliance on our technology. While the goal of our work is to aid the detection of PD, it should not replace the professional diagnosis of healthcare providers. Misinterpretation or misuse of this technology may lead to false positives or negatives, causing unnecessary distress or false reassurance. Lastly, issues of inequity may also arise. Access to advanced diagnostic tools such as the one we propose may be limited, due to geographic location, financial constraints, or digital literacy. As such, this technology could inadvertently widen the healthcare disparity between different socioeconomic groups.

In light of these potential societal impacts, it is essential that proper protocols and measures are put in place to guide the ethical use of such technologies. This includes clear communication about the tool's intended use, rigorous validation processes, and ongoing dialogue about equitable access to and use of these technological advances.

### F.3  Mitigating Bias and Negative Societal Impacts

Some ethical risks exist in the originally publicly available YouTube videos. We are aware that we cannot mitigate those risks to zero. There will be a rest risk. Again, we emphasize that our intention is to enhance the better understanding of PD, its effects on facial expressions, facial landmarks, and audios, as well as explore the potential for technological advancements in this field. We aim to ensure the highest possible standards of ethical conduct in downstream research.

### F.4  Responsibility of AI-Assisted Systems

As mentioned in Section 7 of the main paper, our benchmark aims to inspire a PD early screening tool based on modern machine learning and computer vision techniques. This tool would assist primary clinicians in identifying individuals who may be displaying early physical signs indicative of an evolving Parkinson's syndrome. These individuals can then undergo further neurological evaluation as clinically indicated. This proactive approach will expedite diagnosis and treatment, potentially leading to improved outcomes. On the other hand, while we acknowledge the potential of facial videos and audios in aiding PD detection, we do not advocate for clinicians to rely solely on these modalities for diagnosis. Instead, if positive detection results emerge from facial videos and audios, we recommend that patients seek medical attention at an earlier stage and obtain a more comprehensive diagnosis using additional assessments, such as Dopamine Transporter Scan (DAT), Magnetic Resonance Imaging (MRI), and Positron Emission Tomography (PET).

