# YouTubePD: A Multimodal Benchmark for Parkinson's Disease Analysis Supplementary Material

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

facial expression videos contain. This strategy also allows for a finer classification. In addition, we do not apply UPDRS because facial expression and audio have distinct UPDRS standards. We instead use the holistic severity and confidence annotation based on the video. Doing so ensures the label consistency between the audio data and other modalities, thereby facilitating multimodal PD classification.

In addition, we provide (i) illustrations of our facial landmarks and regions in detail in Figures A1, A2, and A3; (ii) the longitudinal statistics of our PD videos showing their distribution in time in Figure A4.

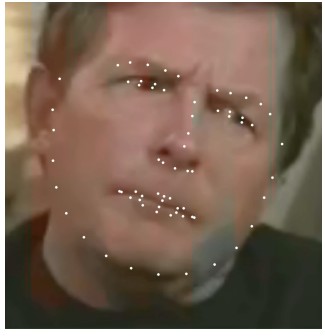

Figure A1: Original landmark extraction.

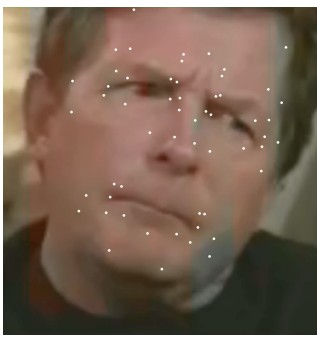

Figure A2: Interpolated landmarks.

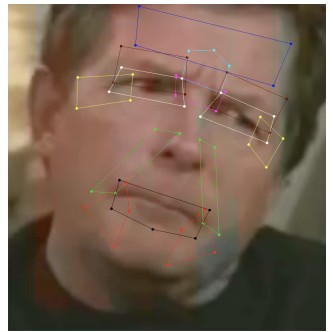

Figure A3: Visualized regions from interpolated landmarks.

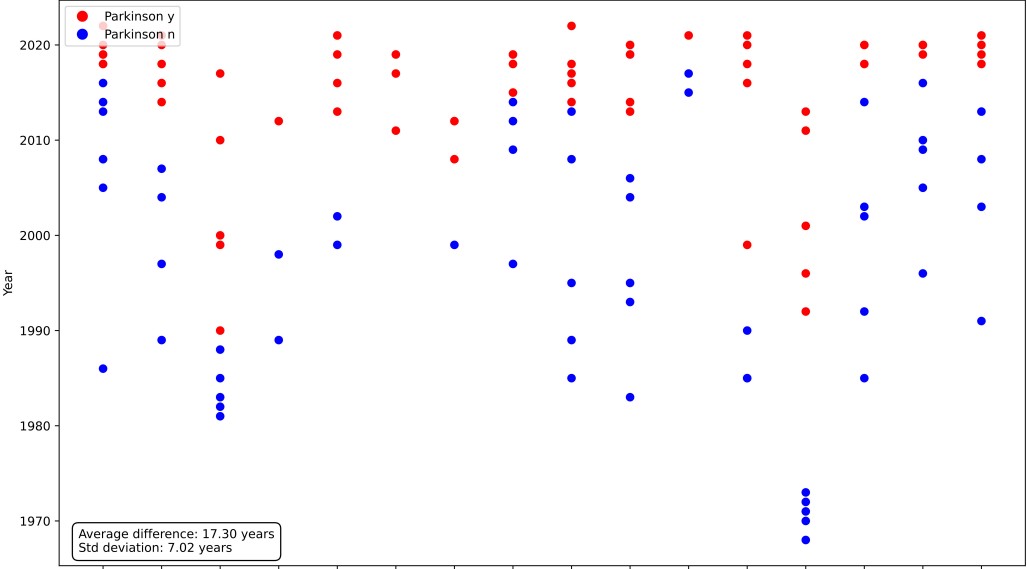

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

| W2V [1] | 57.90($\pm$7.46) | 0.14($\pm$0.02) | 0.43($\pm$0.04) | 4.50($\pm$0.93) |
| MFCC [2] | 39.42($\pm$9.66) | 0.11($\pm$0.03) | 0.49($\pm$0.06) | 6.48($\pm$1.80) |

Table A4: Ablation study on audio representations for multiclass classification on YouTubePD. 'MAE' denotes masked auto-encoders presented in the main paper; 'W2V' denotes wave2vec 2.0; 'MFCC' denotes MEL-frequency cepstral coefficients. We find that MAE provides the most stable and consistent results – we prioritize F1 and AUROC, as the other metrics are influenced by the data imbalance.

**Multimodal fusion strategies.** We conduct an ablation study on the multimodal fusion strategy. Specifically, we explore two different strategies to produce the logits for the facial expression video and facial landmark modalities. One way is frame concatenation (FC) where we concatenate the frame features, and the other is frame voting (FV) where we perform voting to aggregate the result of each frame. Note that FV is used for results in Table 3. For FC, we concatenate frames' features to one vector representing the whole video (either image features from ResNet or landmark coordinates). Then, we train a video-level classifier to obtain the video logits. For FV, we train a frame-level classifier for each frame and average the predictions as video-level logits. For both strategies, the video-level logits from different modalities are averaged to get the final prediction. As shown in Table A5, with FC, the multimodal performance is even lower than the single facial expression modality on F1 and AUROC metrics. By contrast, the FV strategy helps to improve performance.

| Audio Feature | Top-1 Acc ↑ | F1 ↑ | AUROC ↑ | MSE ↓ |
|---|---|---|---|---|
| VGGFace [5] | 78.20($\pm$3.13) | 0.23($\pm$0.02) | 0.68($\pm$0.01) | 2.29($\pm$0.77) |
| Multimodal (FC) | 79.23($\pm$1.94) | 0.21($\pm$0.02) | 0.69($\pm$0.01) | 1.76($\pm$0.18) |
| Multimodal (FV) | 82.75($\pm$2.85) | 0.28($\pm$0.02) | 0.80($\pm$0.03) | 1.40($\pm$0.25) |