# OpenReview forum: "YouTubePD: A Multimodal Benchmark for Parkinson’s Disease Analysis"
_NeurIPS.cc/2023/Track/Datasets_and_Benchmarks — NeurIPS 2023 Datasets and Benchmarks Poster_

### Official Review · Reviewer_tFQT · 2023-07-14
**The authors have come up with the YouTubePD dataset for the diagnosis of Parkinson’s Disease (PD). The dataset consists of video, audio, and detected facial landmarks annotations. The video includes people who are public figures as well. There three tasks, single PD classification, multimodal classification, and PD progression synthesis over time. Deep learning models have been trained and tested and various performance measures are reported in the paper.**

**Rating:** 8
**Confidence:** 5
**Correctness:** Yes
**Clarity:** Yes

**Strengths:**

The paper is well written.
The subject of the paper is a serious social issue.
The dataset is really a great help to the research and medical community.
Good experiments with deep learning.

**Additional Feedback:**

My comments are as follows.
1. I highly respect for such a strategy. You earned it.
2. Are these subjects from around the globe, or majority are from some specific countries, if you have noticed.
3. IS there only one expert?
4. line 94, It would be nice to mention some tasks already here.
5. line 104, What is the distribution of their land of origin? Any comments? I was able to know some of the people, but they are only a few. Because if there is diversity in the land, their facial attributes may be significantly different regardless of PD or NOT.
6. line 111-112, Have you done some video enhancements before cropping/resizing? e.g., there may be some poor contrasts, and so on..
7. line 118-119, Are these landmarks generated:
a. Fully by machine learning algorithms
b. Fully by the human expert(s)
c. First by machine and then corrected by the human expert(s).

**Documentation:**

Sufficient details are given.

**Ethics:**

There are some ethical issues, I am not an expert, though.
. Even though the videos are public, the consent taken from the people in the videos is not clear.

**Limitations:**

Dataset size on a major part, is the issue as far as model generalization is concerned.

**Opportunities For Improvement:**

The dataset size is too small. However, I greatly appreciate the efforts.
My comments are as follows.
1. I highly respect for such a strategy. You earned it.
2. Are these subjects from around the globe, or majority are from some specific countries, if you have noticed.
3. IS there only one expert?
4. line 94, It would be nice to mention some tasks already here.
5. line 104, What is the distribution of their land of origin? Any comments? I was able to know some of the people, but they are only a few. Because if there is diversity in the land, their facial attributes may be significantly different regardless of PD or NOT.
6. line 111-112, Have you done some video enhancements before cropping/resizing? e.g., there may be some poor contrasts, and so on..
7. line 118-119, Are these landmarks generated:
a. Fully by machine learning algorithms
b. Fully by the human expert(s)
c. First by machine and then corrected by the human expert(s).

**Relation To Prior Work:**

Yes

**Summary And Contributions:**

The authors have come up with the YouTubePD dataset for the diagnosis of Parkinson’s Disease (PD). The dataset consists of video, audio, and detected facial landmarks annotations. The video includes people who are public figures as well. There are three tasks, single PD classification, multimodal classification, and PD progression synthesis over time. Deep learning models have been trained and tested, and various performance measures are reported in the paper.

The contributions can be seen as
1. A dataset for PD classification
         a. with single video-based modality
         b. with multimodality
2. PD progression synthesis
         a. with single video-based modality
         b. with multimodality
3. Experiments are done

---

> ### Author Response · Authors · 2023-08-29
> **Author Response**
>
> We appreciate the detailed feedback you provided for our submission. We are encouraged by your acknowledgement that “the subject of the paper is a serious social issue”, “dataset is really a great help”, we accomplish “good experiments with deep learning”, etc. We provide the following clarifications in response to your concerns and have revised the submission accordingly:
>
> **The dataset size is small**
>
> First, we would like to clarify that our dataset is comparable in size to existing datasets in the field of Parkinson’s Disease (PD) analysis, yet it stands out as the first dataset to be publicly available and featuring multiple modalities. As shown in Table 1 of the main paper, our dataset has a comparable number of videos to the previous datasets. Our collection of 283 videos positions it as the second-largest, following only Bandini et al. [4] which has 306 videos, and surpassing other datasets that typically contain around 100 to 200 videos. The constraint on the dataset size primarily comes from the availability of publicly accessible videos: the available PD data are inherently limited.
>
> Second, we would like to emphasize that our benchmark was constructed with the intent of developing novel techniques that deal with the inherent challenge of limited data in the PD analysis problem. For example, our proposed baseline method addresses the data scarcity in ways of leveraging transfer learning from related domains, exploiting informative regions and additional clues in non-PD subjects, and further incorporating multiple modalities. For the PD progression synthesis task, our dataset can be utilized as a challenging testbed to benchmark generative models in few-shot regimes – Table 4 of the main paper evaluates different strategies, like pre-training and fine-tuning and few-shot GAN models.
>
> Third, empirically, models trained on our dataset generalize to real-world clinical PD data as shown in Table 5 of the main paper. Our method achieves an F1 score that is 10% and 16% higher than the baseline on two different splits of real patient data, respectively. These results indicate that the size of our dataset does not hinder the model’s generalization ability.
>
> Finally, the dataset size can be further expanded by including additional subjects and more videos. We thank the reviewer for recognizing our efforts and strategy. In collecting our dataset, we have meticulously included almost all North America public figures that have a public record of PD diagnosis, along with a substantial collection of their videos on YouTube. Beyond providing this dataset, our contribution also includes *establishing an important protocol/strategy* for collecting a PD dataset. Therefore, our dataset could be easily extended in the future with more public figures or individuals from additional geological regions and different social media platforms other than YouTube. We hope our work could inspire efforts to generate larger PD benchmarks.
>
> **Diversity of the dataset;  Distribution of the public figures' land of origin; Line 104**
>
> Statistics of the PD positive group:
>
> | Country        | Count | Race                  | Count | Gender | Count |
> |----------------|-------|-----------------------|-------|--------|-------|
> | United States  | 12    | White/Caucasian       | 13    | Male   | 15    |
> | United Kingdom | 3     | Black/African descent | 3     | Female | 1     |
> | Canada         | 1     |

---

> > ### Comment · Reviewer_tFQT · 2023-08-29
> >
> > Thank you for the explanation

---

> ### Author Response · Authors · 2023-08-29
> **Author Response, Cont'd**
>
> Statistics of the Healthy Control (HC) group:
>
> | country        | count | race                           | count | gender | count |
> |----------------|-------|--------------------------------|-------|--------|-------|
> | South Africa   | 1     | South African+Swiss-German     | 1     | Male   | 70    |
> | United States  | 54    | Black/African descent+Filipino | 1     | Female | 23    |
> | United Kingdom | 12    | Black/African descent          | 11    | ~      | ~     |
> | Israel         | 3     | White/Caucasian                | 64    | ~      | ~     |
> | Russia/Canada  | 1     | Indian                         | 1     | ~      | ~     |
> | Sweden         | 2     | Latinx                         | 5     | ~      | ~     |
> | Kenya/Mexico   | 2     | Asian                          | 9     | ~      | ~     |
> | Brazil         | 2     | Black/African descent+Samoan   | 1     | ~      | ~     |
> | Serbia         | 1     | ~                              | ~     | ~      | ~     |
> | South Korea    | 2     | ~                              | ~     | ~      | ~     |
> | Puerto Rico    | 1     | ~                              | ~     | ~      | ~     |
> | Mexico         | 1     | ~                              | ~     | ~      | ~     |
> | Japan          | 1     | ~                              | ~     | ~      | ~     |
> | Canada         | 4     | ~                              | ~     | ~      | ~     |
> | Denmark        | 1     | ~                              | ~     | ~      | ~     |
> | Russia         | 2     | ~                              | ~     | ~      | ~     |
> | Germany        | 1     | ~                              | ~     | ~      | ~     |
> | France         | 2     |
>
> In the tables above we show the geological, race and gender diversity in our dataset. We have more diverse samples in the healthy control group, despite the PD data being mostly from North America due to PD data availability on YouTube. Additionally, as mentioned above, our dataset is easily extensible using our proposed protocol. By collecting data from more public figures and different social media, the data diversity could be further improved.
>
> We acknowledge the reviewer’s point that the facial features of subjects from different geological locations may differ significantly. However, their PD symptoms will be similar, for example, masked-like facial expression, tremors, etc. We believe our benchmark and method have the potential to generalize to people from other geological locations and races, and we leave detailed investigations to future work.
>
> **Is there only one expert?**
>
> Yes, we have one expert. However, we would like to point out that in the construction of our dataset, we made use of annotation information beyond what a single annotator provided. Since we did not construct the dataset completely from scratch but repurposed existing YouTube videos, we explicitly utilized the topic of these YouTube videos and their meta-information. Therefore, we precisely know which videos are PD-positive and which ones are negative, and there will very unlikely be mistakes in the annotation of video-level PD labels. In addition, the longitudinal meta-information associated with the YouTube videos provides useful prior knowledge about the PD severity. More importantly, **our expert has proficiency in the PD domain with over 20 years of experience**. Empirically, models trained on our annotated dataset generalize to real-world clinical PD data, which reflects the reliability of our annotation.
>
> **Line 94 revision**
>
> Revised with thanks.
>
> **Line 111-112; Video enhancement**
>
> We used high-quality YouTube interview videos. Therefore, there is no need to apply particular enhancements.
>
> **Line 118-119; How the landmarks are generated**
>
> The landmarks are first generated by the machine and then corrected by the human expert. Please refer to Figures 5, 6, and 7 in the revised paper for visualizations of the landmarks.
>
> **The consent taken from the people in the videos**
>
> In line with ethical norms, we have made efforts to obtain explicit consent from each public figure featured in the videos. We remove videos of public figures who wish not to be part of our dataset – these cases are in fact very rare, demonstrating the support our research efforts receive from public figures in general. Also, it is important to note that the public figures have openly discussed their condition in public forums, indicating their willingness to share their Personal Health Information. Please refer to our more comprehensive responses to the ethics reviews.

---

> > ### Comment · Reviewer_tFQT · 2023-08-29
> >
> > Thank you for the explanation

---

> ### Comment · Reviewer_tFQT · 2023-08-29
>
> Thank you for the explanation

---

### Official Review · Reviewer_xcbn · 2023-07-21
**review result**

**Rating:** 5
**Confidence:** 3
**Correctness:** basically correct
**Clarity:** it is OK

**Strengths:**

Providing multimodal data (including videos, audio, and facial landmarks) is featured in this research, compared to existing datasets [3,4,17,24,31,39,16].

The authors also discuss the limitations of their work in discussing the experimental results, and we appreciate their honesty.

**Additional Feedback:**

no special feedback

**Documentation:**

basically no problem

**Ethics:**

there is concern about using Youtube data without signing any contract with the persons or channels

**Limitations:**

It would be crucial to assess how the authors dealt with privacy and ethical concerns, considering the sourcing of data from YouTube.

**Opportunities For Improvement:**

experimental result has room for improvement, is that the reason why this work was dropped here

**Relation To Prior Work:**

clearly described

**Summary And Contributions:**

This paper creates a benchmark dataset, YouTubePD, which is lacked most PD research. The dataset is publicly available, enabling more researchers to do this research.

---

> ### Author Response · Authors · 2023-08-29
> **Author Response**
>
> We appreciate the detailed feedback you provided for our submission. We are encouraged by your acknowledgement that our dataset “features multimodal data”, “enabling more researchers to analyze PD”, “appreciates the discussion of limitations”, etc.  We provide the following clarifications in response to your concerns:
>
> **Experimental result has room for improvement**
>
> Our main contribution is the public Parkinson’s Disease (PD) benchmark instead of the baseline method. We argue the key bottleneck in PD analysis is the lack of a public benchmark. With such a benchmark to fairly compare different methods, we expect future work to build upon our baselines and make further improvements.
>
> Please note that our baseline approaches are also non-trivial. We first evaluate existing approaches in computer vision and identify their limitations. We have discovered that without integrating the informative facial region annotation, existing methods may overfit to spurious correlations and generate predictions using wrong cues. For example, existing methods may have their attention focused on the background instead of facial regions when predicting the PD severity.
>
> Therefore, we employed appropriate strategies to integrate the facial region information using a specialized attention mechanism for PD analysis. Our baseline method also includes designs to tackle data imbalance and deal with limited data. Our baseline approaches obtain good performance on both our YouTube benchmark and real clinical data. It also provides a first step for future work to develop more advanced methods.
>
> **Privacy and Ethics**
>
> Please refer to our more comprehensive responses to the ethics reviews. In summary, we unequivocally share the ethical concerns raised by the reviewer. We would like to highlight that the public figures chose to make their struggles with Parkinson's disease public and discussed their disease and diagnosis in front of cameras. By willingly revealing their identifiable faces and voices, the public figures do not intend to keep their PD information fully private. Please note that the PD videos in our benchmark feature public figures discussing their experience with PD in interview settings. We believe that the concern regarding a breach of privacy is not a newly raised issue specific to our work, and the possibility of any misuse of these videos already exists.
>
> The central question we posed to ourselves was whether sharing these videos with our research community, along with annotations of facial expressions, would amplify the risk of misuse. We are of the opinion that this action does not escalate the aforementioned risk. To further address this matter, we took the initiative to contact the public figures involved and requested permission. This step was taken proactively, particularly in the event that the public figures had regrets about their previous decision to go public and now wished to make a different choice.

---

### Official Review · Reviewer_NCw3 · 2023-07-22
**A Multimodal Benchmark for Parkinson’s Disease Analysis**

**Rating:** 5
**Confidence:** 5
**Correctness:** The authors fail to discuss how to co…
**Clarity:** Yes.

**Strengths:**

The benchmark will be publicly available.

**Additional Feedback:**

No.

**Documentation:**

Yes.

**Ethics:**

Yes.

**Limitations:**

The authors have addressed the limitation of having no public availability of benchmarks. But the authors seldom discuss how to propose a good benchmark. Just copying or collecting videos with annotations from professional clinicians is a perfect solution? It seems that the authors should have other design and implementation decisions.

**Opportunities For Improvement:**

The authors have discussed the limitation of the current work. Compared with other benchmarks, the proposed benchmark has no distinguished characteristics besides being publicly available. The authors also fail to discuss the difference between clinical settings and Youtube video settings. If we presume there are abundant PD videos (maybe fewer), what is the effect of different sampling policies on the design and implementation of benchmarks? I like to see a more quantitative evaluation of how decisions in the design and implementation of benchmarks impact the evaluation results.

**Relation To Prior Work:**

Yes.

**Summary And Contributions:**

 This work introduces YouTubePD -- the first publicly available multimodal benchmark designed for PD analysis.
The authors propose three challenging and complementary tasks encompassing both discriminative and generative tasks, along with a comprehensive set of corresponding baselines.

---

> ### Author Response · Authors · 2023-08-29
> **Author Response**
>
> We appreciate the detailed feedback you provided for our submission. We provide the following clarifications in response to your concerns:
>
> **The proposed benchmark has no distinguished characteristics besides being publicly available**
>
> First, we would like to argue that being the **first** publicly available benchmark is in fact a significant contribution in the field of Parkinson’s Disease (PD) analysis. We highlight such significance from two aspects as follows:
>
> (1) The importance of public datasets and benchmarks in various areas of machine learning is self-evident. Public benchmarks in machine learning provide a standardized platform, which enables consistent and fair evaluation and comparison of algorithms, highlights gaps in current techniques, and drives the development of more powerful models. However, while being instrumental in progressing the field, there has been no such a benchmark in the field of PD analysis, and we have created one. As the reviewers point out, our proposed dataset is “*of interest in order to allow the community to study and fairly compare novel approaches for PD analysis*” (**Reviewer 6pw4**).
>
> (2) Our work signifies a paradigm shift in the establishment of benchmarks in the field of PD analysis, and potentially in healthcare in general. In existing benchmarks, efforts are made primarily for collecting clinical data (i.e., recruiting patients and recording private data), which is labor-intensive and has privacy constraints; therefore, the resulting benchmarks are exclusively private and it is very difficult to construct a public benchmark by following this protocol. By contrast, we overcome this issue by introducing a new perspective – repurposing publicly available data such as YouTube videos – and successfully construct a public benchmark. We hope our paradigm and protocol can be leveraged to facilitate other domains in healthcare (e.g., Alzheimer's Disease analysis) that encounter similar challenges in devising public benchmarks.
>
> Second, in addition to being publically available, our benchmark features other important unique characteristics, as recognized by other reviewers. (1) Our dataset provides “*multimodal data (including videos, audio, and facial landmarks)*” (**Reviewer xcbn**). The complementary information from different modalities is shown to be greatly helpful in PD analysis, which is largely neglected in previous work. Therefore, our dataset can also serve the purpose of benchmarking multimodal learning techniques. (2) We propose a “*novel and interesting PD progression synthesis task*” (**Reviewer 6pw4**). This task provides a novel perspective for PD analysis research via the generative modeling approaches. On the other hand, our dataset can be also utilized as a challenging testbed to benchmark generative models. In summary, our dataset is “*really a great help to the research and medical community*” (**Reviewer tFQT**).
>
> **Difference between clinical settings and YouTube video settings**
>
> We acknowledge there are differences between the clinical setting and our YouTube video setting. One main difference is that the clinical PD videos are often collected in a more controllable manner, where the patients could be instructed to perform specific actions or facial movements. Such control in the clinical setting could simplify PD analysis. By contrast, our YouTube videos are free-form, and there are much less restrictions over the expression and speech of the person. Therefore, analyzing PD from YouTube facial videos is more difficult.
>
> Another difference is that in PD analysis, clinical data are difficult to collect and have privacy constraints. Typically, these clinical data cannot be published due to these privacy concerns. We provide a new perspective of public YouTube data, aiming to provide a public benchmark to fairly compare different approaches for PD analysis.
>
> In Table 5, we show the model trained on our dataset generalizes to real patient data collected in the clinical setting. This result (1) validates that our dataset does not have a significant domain gap to real clinical data, so it is useful in the clinical setting; (2) further confirms our benchmark is meaningful in fulfilling its purpose of publicly and fairly evaluating different approaches for PD analysis in the clinical setting.

---

> ### Author Response · Authors · 2023-08-29
> **Author Response, Cont'd**
>
> **If we presume there are abundant PD videos, what is the effect of different sampling policies on the design and implementation of benchmarks? More quantitative evaluation**
>
> We would like to first kindly point out that the reviewer’s assumption that “there are abundant PD videos” does not hold in practice. In fact, one of the key challenges in constructing a PD benchmark lies in the inherent scarcity of publicly available PD videos. We have meticulously included almost all public figures that have a public record of PD to build our current dataset. Therefore, evaluating the effect of different sampling policies, as suggested by the reviewer, is practically difficult.
>
> Importantly, we have carefully designed the selection and processing protocol for these videos. We start with a list of public figures that have a public record of PD. Then we search for YouTube videos of them spanning a diverse and balanced time period. This is followed by our manual extraction of short clips from long-form videos that clearly show the PD symptoms. In this process, we remove the parts with ambiguity, occlusion, and bad video quality. We also carefully selected similar videos of public figures with non-PD as a control group, in order to avoid the domain gap between PD and non-PD samples. Finally, our clinical expert annotates the videos to further examine and validate the reliability of the data and labels.
>
> We have also validated the design and implementation of our benchmark. In Table 5, we show the model trained on our dataset generalizes to real patient data collected in the clinical setting. This further validates that our benchmark is meaningful in fulfilling its purpose of publicly and fairly evaluating different approaches for PD analysis in the clinical setting. We would kindly request the reviewer to provide specific questions regarding more quantitative analysis to validate our benchmark, and we are more than happy to conduct.
>
>
> **Discussion on how to propose a good benchmark**
>
> We thank the reviewer for the comment. We respectfully disagree with the reviewer that our work is just copying or collecting videos with annotations from professional clinicians. Instead, we believe that our work signifies a paradigm shift in establishing benchmarks in a principled way in the field of PD analysis, which could serve as a valuable exemplar for other domains in healthcare facing similar challenges in devising standardized benchmarks. More specifically, as clarified above, (1) we introduce a novel perspective by repurposing publicly available data, diverging from the conventional practice of collecting clinical data, so different PD analysis methods could be fairly compared under our public benchmark; (2) our benchmark features multiple modalities, which is neglected in prior work; (3) our benchmark encompasses both discriminative and novel generative tasks; (4) the construction of our benchmark involves important implementation decisions and meticulously conducted processing procedures.
>
> While we do not claim to be the best possible PD benchmark, we believe that our work makes an important first step in this direction. We present a complete and well-designed protocol of building such a public benchmark that future work could build on. We also empirically show our current benchmark generalizes to the clinical setting (Table 5) due to our careful design of balanced, diverse sampling. This generalization evaluation further affirms that we have proposed a good benchmark with sound design and implementation decisions.

---

### Official Review · Reviewer_xYDp · 2023-07-23
**Comments**

**Rating:** 6
**Confidence:** 3
**Clarity:** Well written.

**Strengths:**

-  The contribution of this PD dataset is quite big for medical-related research about the PD patient facial expression study.
- Three tasks are proposed with baseline results for the following research.
- Provide experiment splitting.


**Additional Feedback:**

See above.

**Correctness:**

- In the experimental results (Top-1 Acc), the binary accuracy of the multimodal model (ours 71.24) is worse than the multi-class (80.84) which is not consistent with other experimental results. And equipped with multi-modal (71.24) performs worse than only employing VGGface (83.56).


**Documentation:**

- For the annotation of the PD datasets, the current annotation is generated by a human clinician with severity label and confidence level and others. However, most other medical-related datasets, usually employ several clinicians (with many years of professional experience) to diagnose the same patient since different clinicians may generate diverse diagnosis results with different confident levels. And all the following analyses are based on this diagnosis results, it would be better to ensure the label accuracy.
- The dataset contains both data before and after PD is diagnosed. But it is not clear how long the time gap is there. For each patient, do you have the longitudinal data with the same time gap for all the patients?


**Ethics:**

- I would appreciate it if the authors provided the necessary information about the copyright, privacy, and ethics issues regarding the dataset since the dataset containing high-resolution facial images aims to provide public usage for a wide range of research purposes.

**Limitations:**

- The clinical diagnosis of PD patients may not be that reliable. (See my following detailed comments.)
- The data split is based on the video level. But each patient has multiple videos, which means that the same patient may occur both on the training and testing splits.
- The dataset description needs more illustration. For example, the 14 facial regions and extracted 104 landmarks are not clear.
- The size of the dataset is small. Especially for the PD progression Synthesis task, only 11 patients were for training and 5 for testing.


**Opportunities For Improvement:**

- The clinical impact is not clear or not highlighted. I would suggest the authors provide more information about the clinical impact either for patients or for the clinician treatment assistant.
- More background is needed. The diagnosis of PD is not only according to the facial or audio information. Why do the authors focus on the facial and audio information instead of the medical-related data like DAT, MRI, PET？
- More statistical information on the dataset should be provided. Like class numbers and correlations.
- Clinical background is missing.
- Missing illustration on the landmark annotations.



**Relation To Prior Work:**

N/A

**Summary And Contributions:**

This paper describes a new open-access dataset for Parkinson’s disease (PD) with multi-modality data (video, audio, and landmark) from Youtube. It contains 283 videos from 16 PD subjects and 89 Healthy controls. Three tasks and benchmark baselines are designed Facial-expression-based PD classification, multimodal PD classification, and PD progression synthesis.

---

> ### Author Response · Authors · 2023-08-29
> **Author Response**
>
> We appreciate the detailed feedback you provided for our submission. We are encouraged by your acknowledgement that “the contribution of this PD dataset is quite big”, “provide baseline and experiment splitting”, etc. We provide the following clarifications in response to your concerns:
>
> **Clinical impact; More background**
>
> The long-term goal of our work is to develop a Parkinson’s Disease (PD) early screening tool to aid primary clinicians in the earlier recognition of persons exhibiting physical signs that may indicate an evolving Parkinson’s syndrome. Those persons may then undergo additional neurological evaluation as clinically indicated. This will help patients to be diagnosed and treated sooner. Many patients with Parkinson's, in retrospect, exhibited subtle signs of Parkinson’s for months or even years. The signs were not recognized by the patient, their family, or primary clinicians and the signs instead were attributed to aging, and diagnosis was delayed. We hope our effort could eventually inspire such a PD early screening tool to detect PD symptoms at an early stage, and the patients could get timely treatments.
>
> **The diagnosis of PD is not only according to the facial or audio information; DAT, MRI, PET**
>
> The tool is not for Parkinson’s syndrome diagnosis in isolation. The diagnosis of PD is by clinical history in conjunction with clinical examination and at times supplemented by ancillary testing (DaTScan, for example). Facial and audio features were focused on as the main use case for the tool as a screening device (recording and analyses of visual/audio features easily scalable at the primary care level).
>
> Despite our positive experimental results, we do agree with the reviewer that the diagnosis of PD patients may not be completely reliable with solely facial or audio information. Some subjects may not have idiopathic Parkinson’s disease but instead a Parkinson’s variant (multiple system atrophy, diffuse Lewy body disease, progressive supranuclear palsy, drug-induced, vascular, etc.) or be experiencing other conditions that affect facial activation patterns. However, our approach mainly aims to inspire an early detection tool that could be massively deployed with low cost. By using facial videos that could be easily collected using webcam or smart phones, we could provide valuable early detection of PD symptoms. If such positive detection results appear, the patient could seek medical attention at an earlier stage and obtain a more comprehensive diagnosis using these additional information, for example, DAT, MRI, PET.
>
> **More statistical information**
>
> Please note that we have included statistics in the paper. We classify PD in the UPDRS scale. The number of samples per class is listed in the *Videos* portion of Section 3. We investigate the correlation between the region annotations and severity annotation in Section A2.2 in the supplementary material. In addition, we provide country and gender statistics of our dataset subjects in Table 10 of the revision. Please kindly point out what additional statistics would be beneficial, and we are more than happy to include them.
>
> **Illustration on the facial regions and landmarks**
>
> We provide additional illustrations of the off-the-shelf landmark extraction and interpolated keypoints without drawn bounding boxes. The original, off-the-shelf facial keypoint extraction outputs are shown in Figure 5 of the revision. The 104 interpolated keypoints for our method are shown in Figure 6 of the revision. Each index of the interpolated keypoints is mapped to one of the 14 informative facial regions, representing bounding box coordinates for each region. The regions and bounding boxes are visualized in Figure 7 of the revision by connecting the associated interpolated landmarks. We would like to note that example illustrations are included in the dataset repository.

---

> ### Author Response · Authors · 2023-08-29
> **Author Response (2)**
>
> **The size of the dataset is small**
>
> First, we would like to clarify that our dataset is comparable in size to existing datasets in the field of Parkinson’s Disease (PD) analysis, yet it stands out as the first dataset to be publicly available and featuring multiple modalities. As shown in Table 1 of the main paper, our dataset has a comparable number of videos to the previous datasets. Our collection of 283 videos positions it as the second-largest, following only Bandini et al. [4] which has 306 videos, and surpassing other datasets that typically contain around 100 to 200 videos. The constraint on the dataset size primarily comes from the availability of publicly accessible videos: the available PD data are inherently limited.
>
> Second, we would like to emphasize that our benchmark was constructed with the intent of developing novel techniques that deal with the inherent challenge of limited data in the PD analysis problem. For example, for the PD progression synthesis task, our dataset can be utilized as a challenging testbed to benchmark generative models in few-shot regimes. Table 4 of the main paper explicitly evaluates different strategies, like pre-training and fine-tuning and few-shot GAN models. As mentioned in Line 294-296, our comparisons reveal the substantial limitations of state-of-the-art generative models in extracting very fine-grained and subtle information for image translation, further highlighting the importance of our benchmark.
>
> Finally, the dataset size can be further expanded by including additional subjects and more videos. In fact, beyond providing the dataset, our contribution includes *establishing an important protocol* for collecting a PD dataset. Therefore, our dataset could be easily extended in the future with more public figures or individuals from additional geological regions and different social media platforms other than YouTube. We hope our work could inspire efforts to generate larger PD benchmarks.
>
> **Annotation is generated by a human clinician**
>
> We thank the reviewer for the comment on employing several clinicians. We would like to point out that in the construction of our dataset, we made use of annotation information beyond what a single annotator provided. Since we did not construct the dataset completely from scratch but repurposed existing YouTube videos, we explicitly utilized the topic of these YouTube videos and their meta-information. Therefore, we precisely know which videos are PD-positive and which ones are negative, and there will very unlikely be mistakes in the annotation of video-level PD labels. In addition, the longitudinal meta-information associated with the YouTube videos provides useful prior knowledge about the PD severity. More importantly, **our expert has proficiency in the PD domain with over 20 years of experience**.
>
> Empirically, models trained on our annotated dataset generalize to real-world clinical PD data as shown in Table 5 of the main paper. Our method achieves an F1 score that is 10% and 16% higher than the baseline on two different splits of real patient data, respectively. These results reflect that our annotation should be reliable.
>
> **The data split is based on the video level. But each patient has multiple videos, which means that the same patient may occur both on the training and testing splits.**
>
> In our original PD classification setting, our design choice was made to ensure a relatively sufficient number of test data, thus enabling us to generate statistically more reliable results and comparisons.
>
> Additionally, we did evaluate the models trained on our YouTube videos using **never-before-seen PD subjects** from real-world clinical PD data, as shown in Table 5 of the main paper.
>
> As suggested by Reviewer 6pw4, here we evaluate a leave-one-subject-out setting. In the table below, we present the results when we train the facial expression model on a modified training and test split where we ensure there is a PD-positive patient only represented in the test split. We report the average multi-class results of three such leave-one-out splits.
>
> | Model         | F1   | AUROC | Top-1 Acc |
> |---------------|------|-------|-----------|
> | VGGFace | 0.24 | 0.66  | 79.4      |
> | Ours          | 0.25 | 0.71  | 76.4      |
>
>
> In these new splits, we still observe that our method outperforms the baseline under the F1 and AUROC metrics. On the leave-one-subject-out setting, the AUROC of our method improves by 0.05 and F1 improves by 0.01. This improvement is similar to our original setting (AUROC by 0.06 and F1 by 0.02). Please note that similar to the results in the main paper, the Top-1 accuracy is not indicative of model performance, because of the presence of a large amount of negative samples.

---

> ### Author Response · Authors · 2023-08-29
> **Author Response (3)**
>
> **Time gap for all the patients**
>
> We provide the longitudinal statistics in Figure 8 of the revision. There is an average gap of 17.3 years between positive and negative samples of the same public figure.
>
>
>
>
> **Explanation on experimental results**
>
> The binary accuracy and multi-class top-1 accuracy cannot be directly compared. In line 153, we mentioned that though accuracy indicates the performance of the model in the binary setting, top-1 accuracy is not a reliable metric in the multiclass setting. This is because the data is naturally imbalanced and we have a much larger number of negative samples. Please note that under other metrics, the result is consistent along the line of the reviewer's argument.
>
> **Ethics and privacy**
>
> Please refer to our more comprehensive responses to the ethics reviews. In summary, we unequivocally share the ethical concerns raised by the reviewer. We would like to highlight that the public figures chose to make their struggles with Parkinson's disease public and discussed their disease and diagnosis in front of cameras. By willingly revealing their identifiable faces and voices, the public figures do not intend to keep their PD information fully private. Please note that the PD videos in our benchmark commonly feature public figures discussing their experience with PD in interview settings. We believe that the concern regarding a breach of privacy is not a newly raised issue specific to our work, and the possibility of any misuse of these videos already exists.
>
> The central question we posed to ourselves was whether sharing these videos with our research community, along with annotations of facial expressions, would amplify the risk of misuse. We are of the opinion that this action does not escalate the aforementioned risk. To further address this matter, we took the initiative to contact the public figures involved and requested permission. This step was taken proactively, particularly in the event that the public figures had regrets about their previous decision to go public and now wished to make a different choice.

---

### Official Review · Reviewer_6pw4 · 2023-07-23
**A new dataset for Parkinson's Disease analysis**

**Rating:** 8
**Confidence:** 3
**Clarity:** The paper is very clear

**Strengths:**

- The paper is well organized and written
- The acquisition protocol as well as the experiments carried out are well described
- The proposed dataset is of interest in order to allow the community to study and fairly compare novel approaches for PD analysis
- The PD Progression synthesis challenge is very interesting
- Experiments carried out validate that the proposed dataset can be used for real-world applications
- The dataset is publicly available


**Additional Feedback:**



**Correctness:**

Claims seem correct to me. However another protocol for the PD classification could be better to avoid the bias of having the same subject for both training and test

**Documentation:**

well documented

**Ethics:**

no ethical concerns

**Limitations:**

Yes, limitations are discussed


**Opportunities For Improvement:**

- The datasets size is relatively small with only 283 videos from only 16 subjects
- It seems that annotations are provided by only one expert. This may also be an important bias
- For PD classification, it seems that the same subject can be in two different folds and then used at the same time for training and test. This may add a bias in the evaluation of baselines. A leave-one-subject-out protocol may be more appropriate to fairly evaluate and compare the baselines for the PD classification task.

**Relation To Prior Work:**

A clear discussion of previous contributions and how this dataset differs from existing ones is present

**Summary And Contributions:**

In this paper, authors propose a new multimodal dataset for Parkinson’s Disease (PD) analysis: YoutubePD. The dataset is collected from Youtube videos of public persons experiencing Parkinson’s Disease. In total 283 videos of 16 different subjects are gathered with different modalities such as facial expression videos, facial landmarks and audio. Each video is annotated by a clinician expert with detailed annotations including overall severity label and facial region particularly informative. Along with the dataset, authors also propose three tasks: Facial-Expression-Based PD Classification, Multimodal PD Classification and PD Progression synthesis. For each task, several existing approaches and baselines are assessed and compared. Finally, authors experimentally evaluated how models trained on the YoutubePD dataset generalize on real-world data (from a private dataset). Results suggest that the YoutubePD dataset can be used for real-world applications. The proposed datasets will be released publicly.

---

> ### Author Response · Authors · 2023-08-29
> **Author Response**
>
> We appreciate the detailed feedback you provided for our submission. We are encouraged by your acknowledgement that our “proposed dataset is of interest”, “paper is well organized and written”, and “PD Progression synthesis challenge is very interesting”. We provide the following clarifications in response to your concerns:
>
> **The dataset size is relatively small**
>
> First, we would like to clarify that our dataset is comparable in size to existing datasets in the field of Parkinson’s Disease (PD) analysis, yet it stands out as the first dataset to be publicly available and featuring multiple modalities. As shown in Table 1 of the main paper, our dataset has a comparable number of videos to the previous datasets. Our collection of 283 videos positions it as the second-largest, following only Bandini et al. [4] which has 306 videos, and surpassing other datasets that typically contain around 100 to 200 videos. Regarding the number of PD subjects, as we discussed in the limitation section, the constraint primarily comes from the availability of publicly accessible videos: the available PD data are inherently limited. In collecting our dataset, we have meticulously included almost all North America public figures that have a public record of being diagnosed with PD, along with a substantial collection of their videos on YouTube. Meanwhile, we enriched the subject pool by incorporating non-PD subjects, resulting in 200+ subjects in total.
>
> Second, we would like to emphasize that our benchmark was constructed with the intent of developing novel techniques that deal with the inherent challenge of limited data in the PD analysis problem. For example, our proposed baseline method addresses the data scarcity in ways of leveraging transfer learning from related domains, exploiting informative regions and additional clues in non-PD subjects, and further incorporating multiple modalities. For the PD progression synthesis task, our dataset can be utilized as a challenging testbed to benchmark generative models in few-shot regimes – Table 4 of the main paper evaluates different strategies, like pre-training and fine-tuning and few-shot GAN models.
>
> Finally, we agree with the reviewer that the scale of the PD benchmark can be further expanded by including additional subjects and more videos. In fact, beyond providing the dataset, our contribution includes *establishing an important protocol* for collecting a PD dataset. Therefore, our dataset could be easily extended in the future with more public figures or individuals from additional geological regions and different social media platforms other than YouTube. We hope our work could inspire efforts to generate larger PD benchmarks.
>
> **Annotations are provided by only one expert**
>
> We thank the reviewer for raising the potential annotation issue associated with a single expert. We would like to point out that in the construction of our dataset, we made use of annotation information beyond what a single annotator provided. Since we did not construct the dataset completely from scratch but repurposed existing YouTube videos, we explicitly utilized the topic of these YouTube videos and their meta-information. Therefore, we precisely know which videos are PD-positive and which ones are negative, and there will very unlikely be mistakes in the annotation of video-level PD labels. In addition, the longitudinal meta-information associated with the YouTube videos provides useful prior knowledge about the PD severity. More importantly, **our expert has proficiency in the PD domain with over 20 years of experience**.
>
> Empirically, models trained on our annotated dataset generalize to real-world clinical PD data as shown in Table 5 of the main paper. Our method achieves an F1 score that is 10% and 16% higher than the baseline on two different splits of real patient data, respectively. These results reflect that our annotation should be reliable.

---

> ### Author Response · Authors · 2023-08-29
> **Author Response, Cont'd**
>
> **Same patient may occur both on the training and testing splits; Leave-one-subject-out experiments**
>
> We thank the reviewer for suggesting the leave-one-subject-out experiment. In our original PD classification setting, we did not adopt the leave-one-subject-out configuration – this design choice was made to ensure a relatively sufficient number of test data, thus enabling us to generate statistically more reliable results and comparisons.
>
> Additionally, we did evaluate the models trained on our YouTube videos using **never-before-seen PD subjects** from real-world clinical PD data, as shown in Table 5 of the main paper.
>
> Per the reviewer’s suggestion, in the table below, we present the results when we train the facial expression model on a modified training and test split where we ensure there is a PD-positive patient only represented in the test split. We report the average multi-class results of three such leave-one-out splits.
>
> | Model         | F1   | AUROC | Top-1 Acc |
> |---------------|------|-------|-----------|
> | VGGFace | 0.24 | 0.66  | 79.4      |
> | Ours          | 0.25 | 0.71  | 76.4      |
>
>
> In these new splits, we still observe that our method outperforms the baseline under the F1 and AUROC metrics. In the leave-one-subject-out setting, the AUROC of our method improves by 0.05 and F1 improves by 0.01. This improvement is similar to our original setting (AUROC by 0.06 and F1 by 0.02). Please note that similar to the results in the main paper, the Top-1 accuracy is not indicative of model performance, because of the presence of a large amount of negative samples.

---

### Author Response · Authors · 2023-08-29
**General Response**

We thank the reviewers for the valuable comments. Our proposed public Parkinson’s Disease (PD) analysis dataset is *of interest in order to allow the community to study and fairly compare novel approaches for PD analysis* (Reviewer **6pw4**). The contribution of this PD dataset is *quite big for medical-related research about the PD patient facial expression study* (Reviewer **xYDp**).
The dataset is *really a great help to the research and medical community* (Reviewer **tFQT**). Our paper is *well written*(Reviewer **tFQT**, Reviewer **6pw4**, Reviewer **xYDp**) and our PD Progression synthesis challenge is *very interesting* (Reviewer **6pw4**).

To address specific concerns raised by the reviewers, we provide additional clarifications to each question. We also provide additional results and visualizations in the newly added Section 8 “Rebuttal” of the updated manuscript. Specifically, we provide 1) illustrations of our facial landmarks and regions in detail; 2) the longitudinal statistics of our PD videos showing their distribution in time; 3) the results of a leave-one-subject-out analysis, where we experiment on a different train-test split such that the public figures used in testing are unseen in training; 4) the demographic statistics of YouTubePD. Please refer to the individual responses for details.

---

### Decision · Program_Chairs · 2023-09-22

**Decision:**

Accept (Poster)

**Comment:**

This paper offers a good contribution to medical AI. The dataset/benchmark can be used for building different models even using different modalities. There are some ethical concerns that the authors tried to address. They advised to rectify all those concerns for the final version of the paper.